# Automatic Organization of Neural Modules for Enhanced Collaboration in Neural Networks

## Abstract

This work proposes a new perspective on the structure of Neural Networks (NNs). Traditional Neural Networks are typically tree-like structures for convenience, which can be predefined or learned by NAS methods. However, such a structure can not facilitate communications between nodes at the same level or signal transmissions to previous levels. These defects prevent effective collaboration, restricting the capabilities of neural networks. It is well-acknowledged that the biological neural system contains billions of neural units. Their connections are far more complicated than the current NN structure. To enhance the representational ability of neural networks, existing works try to increase the depth of the neural network and introduce more parameters. However, they all have limitations with constrained parameters. In this work, we introduce a synchronous graph-based structure to establish a novel way of organizing the neural units: the Neural Modules. This framework allows any nodes to communicate with each other and encourages neural units to work collectively, demonstrating a departure from the conventional constrained paradigm. Such a structure also provides more candidates for the NAS methods. Furthermore, we also propose an elegant regularization method to organize neural units into multiple independent, balanced neural modules systematically. This would be convenient for handling these neural modules in parallel. Compared to traditional NNs, our method unlocks the potential of NNs from tree-like structures to general graphs and makes NNs be optimized in an almost complete set. Our approach proves adaptable to diverse tasks, offering compatibility across various scenarios. Quantitative experimental results substantiate the potential of our structure, indicating the improvement of NNs.

## 1 Introduction

Neural Networks are often organized hierarchically in a tree-like fashion. However, this conventional approach obstructs effective communication among nodes and the structure can not facilitate information interaction between nodes at the same level or signal transmissions to previous levels. In fact, existing NN structures optimize in a limited space, significantly diminishing the potential of NNs, and impeding their full capabilities. Our work involves a reimagining of traditional NNs. We contend that the existing connectivity approaches inadequately capture the essence of neural networks. The nodes in an asynchronous tree-like structure lack the ability to establish connections, hindering information transfer between neural units and leading to deficiencies. To address this limitation, we introduce a method to build a synchronous graph structure for the nodes by the proposed Neural Modules, fostering collaboration among neural units. This innovative approach aims to overcome the shortcomings of the asynchronous tree-like structure, providing a more effective and interconnected framework for neural networks. Our framework progresses the structure of NN from tree-like structures to general graphs, providing more candidates for NAS methods.

For our framework. the concept of the Neural Module is defined as follows:

**Definition 1.** *A Neural Module is defined as a directed, connected subgraph and the absolute value of the weight for any edge in the subgraph exceeds the predefined threshold $\gamma$.*

$$\gamma = the\ kth\ largest\ absolute\ value\ of\ the\ weights. \tag{1}$$

Our method enables synchronous communication among any nodes within the neural module. This transformative adjustment enhances information transfer, thereby boosting the overall capacity of NN structures. By fostering a collaborative environment among nodes, our approach leverages their collective power to unlock NNs' capabilities. This design promotes collaboration among nodes, facilitating more accurate information transfer. It is noteworthy that the existing tree-like NNs are essentially a substructure of our designed general graph. In our framework, multiple neural units collaboratively execute precise functional implementations. Our innovation is poised to bridge the gap between artificial and more general structures like biological neural networks. Furthermore, our framework provides more candidates for NAS methods.

Designing such an architecture is a challenging task. While some similar works have made efforts, improvement is still needed. We found that the fixed point of the implicitly hidden layer is a solution. Instead of using an infinite structure of hidden layers, we organize it into a graph. However, for a larger number of nodes, the implicitly hidden layer imposes a heavier computational burden and increases the risk of overfitting. For example, the Deep equilibrium models(DEQ) encounter difficulties in finding the fixed point and calculating the Jacobian Matrix as well as its inverse matrix for a larger number of nodes, which may also lead to serious overfitting problems due to poor node organization. To address this, we propose a new regularization method to better organize these nodes into multiple proper neural modules that are independent and can be handled in parallel. This method allows nodes to form an automatic organization, enhancing the overall efficiency of the learning process as well as its performance by reducing overfitting,

Our learning process exhibits adaptability to larger search spaces and diverse tasks, effectively eliminating the structural bias that some nodes cannot transfer information due to the limitations of the current tree-like structure. We evaluate the effectiveness of our optimization method by conducting experiments on state-of-the-art networks, demonstrating its competitiveness compared to existing networks across various real-world tasks and datasets. The results from these experiments indicate the superiority of the learned connectivity in terms of performance and efficiency.

To sum up, our contributions to this study can be outlined as follows: We propose a method to generalize existing tree-like structures to learnable general graphs for NNs and we introduce a novel regularization method to organize the neural units into Neural Modules that help to enhance the efficiency of the structure as well as its performance by reducing overfitting.

## 2 RELATED WORKS

To progress the existing tree-like structure for NNs. Yuan (Kun Yuan & Yan, 2020) recently provided a topological perspective, highlighting the benefits of dense connections offered through shortcuts in optimization (Srivastava et al., 2015) (Sandler et al., 2018). Furthermore, sparsity constraints have also been proven effective in optimizing learned structures across various applications (Srivastava et al., 2015) (Chu et al., 2023) (Ahmed & Torresani, 2018) (He & Sun, 2016) (Huang & Weinberger, 2017). In their approach, the structure of NNs is organized as a DAG, whereas we organize it as a more general graph structure.

The fixed point of the implicitly hidden layer can also be served as a solution (Bai et al., 2019) (Tsuchida & Ong, 2022) (Chu et al., 2023) (Yang et al., 2022) (Heaton et al., 2021) (Zucchet & Sacramento, 2022), as demonstrated in the subsequent works (Bai et al., 2020) (Szekeres & Izsák, 2024) (Yang & Liu, 2023). Departing from the infinite structure of implicitly hidden layers (Chu et al., 2023) (Ling et al., 2023) (Ding et al., 2023) (Yang et al., 2022) (Liu et al., 2022), we organize it into a general graph structure. Furthermore, for larger NNs, we introduce a novel regularization method to organize neural units into multiple independent neural modules. Compared with implicitly hidden layers, our method can improve the efficiency by parallel computing as well as the performance by reducing overfitting. Furthermore, it enhances the interpretability of implicitly hidden layers.

Our process also involves compressing NNs. In recent years, various algorithms have been developed, including quantization (Kai Han & Xu, 2020) (Mingzhu Shen & Wang, 2019) (Yang He & Yang, 2018), low-rank approximation (Li & Shi, 2018) (Zhaohui Yang & Xu, 2019) (Xiyu Yu &

Tao, 2017), knowledge distillation (Shumin Kong & Xu, 2020) (Shan You & Tao, 2018), and network pruning (Pavlo Molchanov & Kautz., 2019), etc. The network pruning method we use in this work is weight pruning, which aims to eliminate weak connections. In addition, the method also involves filter pruning that removes entire redundant filters (Y Tang, 2020). We aim to improve weight pruning for our framework using a method similar to Tao Lin (2020), which evaluates the gradient at the pruned model and applies parameters' updates to the dense model. In our framework, this process coordinates with an elegant regularization to automatically allocate Neural Modules.

OptNet integrates optimization quadratic problems for nodes within the same layer (Amos & Kolter, 2017) (Yan & Zhang, 2021). However, this approach introduces bias and lacks an interpretable structure, affecting overall interpretability. Furthermore, OptNet uses parameters derived from quadratic problems for backpropagation. These parameters, determined by nodes from the previous layer, are difficult to manage in complexity. This complexity not only increases the computational burden but also exacerbates overfitting problems. Using a differentiable function for these parameters adds further bias.

Graph Neural Networks (GNNs) specially address the needs of geometric deep learning (Gori et al., 2005) (Fan et al., 2019) (Scarselli et al., 2008) (Abadal et al., 2021). GNNs adapt their structure to the input graph, capturing complex dependencies (Yong et al., 2007) (Abadal et al., 2021) (Fout et al., 2017) (Fan et al., 2019). This adaptability enables them to predict properties of geometric data, which primarily deal with graphs, our innovation involves evolving the tree-like structure into a general graph format for NNs, expanding their applicability and potential.

The flexibility of this graph structure has also been investigated in studies similar to those on Reservoir Computing (Lukoševicius & Jaeger, 2009) (Benjamin Schrauwen & Campenhout., 2007) (Leshno & Schocken, 1993b). These studies have utilized a recurrent neural network framework where neuron connections are established randomly, where the weights remain static post-initialization. Reservoir Computing capitalizes on the dynamics of a non-linear system. In contrast, our Neural Module framework is designed around a balanced system paradigm, allowing for the adaptive learning of both weights and network structure during processing. This feature enhances our framework's versatility and effectiveness, setting it apart from Reservoir Computing with its stronger capabilities.

## 3 METHODOLOGY

### 3.1 RATIONALE FOR INTRODUCING THE STRUCTURE

NNs represent a type of information flow. However, tree-like structures maintain a hierarchical, nested form that limits information transfer, as each node can only be influenced by its precursor nodes. These structures are asynchronous and lack the flexibility to form cycles. Moreover, we observe that these structures are notably inferior in complexity compared to biological neural networks, which exhibit more intricate connectivity patterns beyond the simplicity of tree-like structures. The adoption of these structures is primarily due to their favorable mathematical properties, particularly in facilitating convenient forward and backward propagation.

Consider the traditional tree-like structure for an NN with $m$ layers. Let $N^0$ be the nodes of the first layer and the input values feed into the first layer. Let $N^m$ be nodes for the last layer and they feed for the output values. For any node in the other layers, $n_i^k \in N^k, 0 < k < m$, it represents the intermediate values. The parameters of the model are represented as the edges in the structure. Let $e_{i,j}^k \in E^k, 1 \le k \le m$ be the the edge connecting $n_i^{k-1}$ and $n_j^k$. Then the traditional tree-like structure for NNs which is organized as an asynchronous hierarchical structure can be formalized as follows: $\mathcal{T} = \{N^0, E^1, N^1, ...E^m, N^m\}$.

In our work, we organize the intermediate structure as a general graph. The model is denoted by $NM$, $\mathcal{NM} = \{N^0, E^1, \mathcal{G}, E^m, N^m\}$, where $\mathcal{G} = \{E, N\}$ and $n_i \in N$ is the $i$th node in $\mathcal{G}$, $e_{i,j} \in E$ is the edge from $n_i$ to $n_j$. Let the number of the nodes in $N$ be $p$, the number of the nodes in $N^0$ be $|N^0|$ and the number of the nodes in $N^m$ be $|N^m|$.

In this way, we would enhance the description of the NNs compared to traditional approaches.

## 3.2 MODEL STRUCTURE

In the traditional forward computation for a tree-like structure, each node in the first layer aggregates inputs and is then transformed by tensor flow. Following the provided definition of NM, our structure is constructed as follows: all the intermediate nodes are organized into a graph, as illustrated in Figure 1. In this arrangement, each node is influenced by all other nodes in the graph. This collective influence enables nodes to work together, significantly improving NNs' characterization.

In our framework, nodes are initially computed based on their input nodes, which solely distribute features. Additionally, each node is influenced by other nodes in the graph, creating mutual influence.

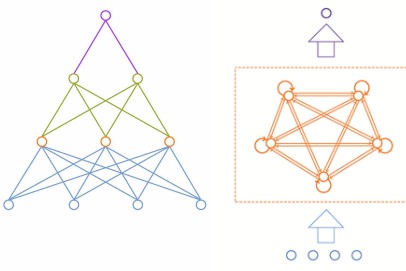

Figure 1: The first image shows the tree-like structure of a traditional NN. In the second depiction, for the intermediate nodes, our model NM introduces a graph. Note that, the nodes' spin part represents its bias for itself.

In the upcoming section, we elaborate on the process of calculating node values using both input nodes and the nodes within the graph $\mathcal{G}$.

## 3.3 FORWARD PROCESS

First, we discuss the case of the value for each node in the graph. Here in this paper, the value for each node is represented as $x$ and the value for each edge is represented as $w$ with the same corresponding index. As introduced in the previous section, these values depend on the nodes in $N^0$ as well as other nodes in $\mathcal{G}$. Hence, we need a synchronization method to address this. We treat the problem as a system of multivariate equations. For the value of the nodes in $\mathcal{G}$, we have the following:

$$
\begin{cases}
w_{1,1} + \sum\limits_{j \neq 1} f(x_j) \cdot w_{j,1} + \sum\limits_{j=1}^{|N^0|} x_j^0 \cdot w_{j,1}^1 = x_1 \\
w_{2,2} + \sum\limits_{j \neq 2} f(x_j) \cdot w_{j,2} + \sum\limits_{j=1}^{|N^0|} x_j^0 \cdot w_{j,2}^1 = x_2 \\
... \\
w_{p,p} + \sum\limits_{j \neq p} f(x_j) \cdot w_{j,p} + \sum\limits_{j=1}^{|N^0|} x_j^0 \cdot w_{j,p}^1 = x_p
\end{cases}
$$

In the above equations, $w_{1,1}$, $w_{2,2}$, ..., $w_{p,p}$ are the values of the self-spin edges for $\mathcal{G}$ and they represent the bias of the nodes. $f$ is the activation function.

Let $W^m$ be the weights of $E^m$ and $X = \{x_1, x_2, ...\}$ be the values of the nodes in $\mathcal{G}$ . Then, according to the definition of $\mathcal{NM}$, the output values $\tilde{Y} = X^m$ can be inferred by $X^m = g(f(X) \cdot W^{mT})$, where $g$ is the activation function for output.

Existing numerical methods such as Newton-Raphson method, can effectively solve these above equations. In real-world applications, besides Newton's methods, the efficiency can be optimized by iterative methods, dichotomy, or secant methods. Note that, each variable is processed by an activation function $f$, making the transformation nonlinear. If we consider $\mathcal{G}$ as a Neural Module,

we can introduce the following theorem. According to Cybenko (1989) Hornik & White (1990) Leshno & Schocken (1993a), the theorem will be proved in the Appendix.

**Theorem 3.1** (Universal Neural Module Approximation Theorem). *Let $\mathcal{F}$ be an implicit function defined on a compact set, capable of being transformed into a continuous explicit function for all the variables. In such a case, there exists a Neural Module that can effectively approximate the function $\mathcal{F}$.*

### 3.4 BACKWARD PROCESS

To determine the gradient of the nodes in the neural module, we consider the gradient of the output as the last layer $\nabla X^m = \nabla Y$. The same as the Forward Process, the nodes' gradients also interact with each other. Note that, each node has been processed by the activation function $f$. Then, we treat the gradient of the nodes in the graph as variables within the following system of equations.

$$
\begin{cases}
\sum\limits_{j \neq 1} \nabla x_j \cdot f'(x_j) \cdot w_{1,j} + \sum\limits_{j=1}^{|N^m|} \nabla x_j^m \cdot g'(x_j^m) \cdot w_{1,j}^m = \nabla x_1 \\
\sum\limits_{j \neq 2} \nabla x_j \cdot f'(x_j) \cdot w_{2,j} + \sum\limits_{j=1}^{|N^m|} \nabla x_j^m \cdot g'(x_j^m) \cdot w_{2,j}^m = \nabla x_2 \\
... \\
\sum\limits_{j \neq m} \nabla x_j \cdot f'(x_j) \cdot w_{p,j} + \sum\limits_{j=1}^{|N^m|} \nabla x_j^m \cdot g'(x_j^m) \cdot w_{p,j}^m = \nabla x_p
\end{cases}
$$

Finally, we can compute the gradient of the edges of our NM. First, for the gradient of the edges in $\mathcal{G}$, according to the system of equations, we need to consider the gradient of each node. For any $j$th node in the graph, the weights of the incoming arc are represented by the $j$th($1 \leq j \leq p$) column of its adjacency matrix. For convenience, we introduce the following operator:

$$
\mathcal{H}_j = [f(x_1), ..., f(x_{j-1}), 1, f(x_{j+1}), ..., f(x_p)] , \tag{2}
$$

This operator is derived from the system of equations in the forward process. Then, by the gradient of the $j$th node, its corresponding gradient for $W_{:,j}^T, 1 \leq j \leq p$ in $\mathcal{G}$ can be formulated as follows:

$$
\nabla W_{:,j}^T = \nabla x_j \cdot f'(x_j) \cdot \mathcal{H}_j . \tag{3}
$$

Second, for the gradient for the edges in $E^m$, according to the forward process,

$$
\nabla W^m = \nabla X^{mT} \cdot g'(X^{mT}) \cdot f(X) . \tag{4}
$$

Third, for the gradient for the edges in $E^1$, according to the forward process,

$$
\nabla W^1 = \nabla X^T \cdot X^0 . \tag{5}
$$

### 3.5 NEURAL MODULE OPTIMIZATION

For convenience, we take $\mathcal{G}$ as a single neural module as stated earlier, this serves as a universal set for all possible connections. In our work, we optimize the structure to prioritize important connections. The optimization process typically involves emphasizing crucial connections in the graph. In this paper, we introduce parameter $\gamma$ acting on the formation of Neural Modules as Definition 1. A larger $\gamma$ generates sparser graphs and results in smaller Neural Modules. Here, we propose a new regularization method to formalize multiple balance Neural Modules, as introduced earlier. And we term it as $NM$ regularization.

The forward and backward process of the structure, as discussed earlier, involves solving a system of equations. Managing a large number of nodes would impose a considerable computational burden when dealing with such systems. However, we can address this challenge by optimizing the graph at any part using $NM$ regularization, transforming the large graph into independent, smaller connected subgraphs that formalize Neural Modules, as previously introduced. As each Neural Module does not connect with others, we can calculate them in parallel. Compared with $L1$ regularization, $NM$ regularization can get more balanced neural modules, which would bring additional benefits for

efficiency as if it is calculated in parallel, the overall efficiency is determined by the largest Neural Module. This approach enables a significant reduction in computation. At the same time, it also promotes the performance by mitigating overfitting.

$NM$ regularization considers the number of edges that have a connected path to each node. Here, we introduce operator:

$$\mathcal{Z} = [z_1, z_2, ..., z_p],\tag{6}$$

For the element $z_k, 1 \leq k \leq p$ in $\mathcal{Z}$, initialized by the connected subgraphs formulated in the first iteration,

$$z_k = the\ number\ of\ edges\ connected\ to\ node\ n_k.\tag{7}$$

Let $\alpha$ be the parameter for regularization. By $NM$ regularization, for the larger connected subgraph, there is an adaptive adjustment to increase $\alpha$ in each iteration. This adjustment results in the automatic organization of the graph into properly balanced subgraphs, forming rational neural modules that effectively utilize neuron units. For the $k$th node in $\mathcal{G}$, $1 \leq k \leq p$, the formulation of our $NM$ regularization is as follows:

$$x_k = w_{k,k} + \sum_{j \neq k} f(x_j) \cdot w_{j,k} + \sum_{j=1}^{|N^0|} x_j^0 \cdot w_{j,k}^1 + \alpha \cdot z_k \cdot \sum_{j \neq k} norm(w_{j,k}),\tag{8}$$

where $norm$ denotes the regular function like $L1$ or $L2$ regularization. In this manner, for larger subgraphs, a larger value of $z$ would cause the edges connected to these subgraphs to tend toward zero, as shown in Figure 2.

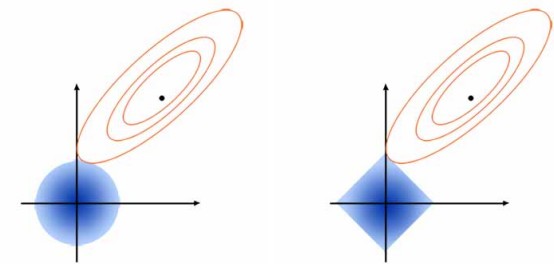

Figure 2: The impact of $L1$ and $L2$ regularization is significant in neural modules. Nodes within larger neural modules have a greater number of edges, which leads to a higher value of $z$ and, consequently, a stronger regularization signal. This results in a more potent force pulling the values towards zero. The parameter $\gamma$ plays a crucial role in pruning edges with smaller weights, which not only prevents the uncontrolled growth of large neural modules but also fosters the emergence of smaller, more manageable neural modules. This is why NM (Neural Module) regularization is instrumental in creating a balanced structure within neural modules.

Based on the outcomes of $L1$ and $L2$ regularization, in the context of the $L1$ norm, the optimal value associated with the node $n_k$ is.

$$w_{j,k}^* = sgn(w_{j,k})(|w_{j.k}| - \alpha \cdot z_k)_+,\tag{9}$$

where $_+$ takes the positive part of the value. For the $L2$ norm case, the best value associated with the node $n_k$ is

$$w_{j,k}^* = \frac{1}{1 + \alpha \cdot z_k} \cdot w_{j,k}.\tag{10}$$

As derived from Equations 9 and 10, an increased value of $z$ leads to the reduction of weights, thereby preventing the growth of large neural modules through the application of the threshold parameter $\gamma$, as previously mentioned. Concurrently, this mechanism fosters the emergence of smaller neural modules by utilizing lower values of $z$. Consequently, NM regularization contributes to the formation of well-balanced, independent neural modules, which, as discussed earlier, offer advantages for parallel computing.

In this method, we only consider the absolute value of the weight for each edge larger than $\gamma$ to solve the system of equations during both the forward and backward processes. With the aid of

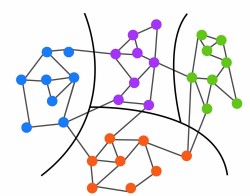

Figure 3: If the node corresponds to a large connected subgraph, $NM$ regularization increases its weight for regularization, making it more prone to tend towards zero. $NM$ regularization opts to divide large neural modules into multiple smaller, independent, balanced ones. This approach confines the calculation for the system of equations to each subgraph, enabling parallelized solutions.

$NM$ regularization, we achieve properly balanced neural modules. Employing this method involves solving large systems of equations into multiple smaller systems of equations. These smaller ones are independent and can be solved in parallel. This strategy enables the framework to handle a large number of nodes efficiently.

Lastly, we refer to the parameter $\gamma$ as an approximation to solve the system of equations in the forward and backward process. This approximation significantly reduces the heaviest computational load. After obtaining the gradient of each node, we calculate the gradient for all edges including those whose absolute values are smaller than $\gamma$, and then update them. This ensures that while $\gamma$ helps to approximate the solution to the system of equations, the update for edges is fully quantified and precise. In this way, our $NM$ regularization achieves higher efficiency as well as better performance and robustness.

## 4 EXPERIMENTS

### 4.1 PERFORMANCE OF NEURAL MODULES

In this section, we present experiments conducted with our Neural Modules, comparing their performance with traditional NN methods, implicit hidden layers (DEQ), a topological perspective treating NN as a DAG, and recently introduced OPTNET. The results are tabulated for three real datasets, all of which are available in the UCI dataset. For the regular function $norm$, we use the absolute value of the wights.

The first dataset comprises codon usage frequencies in genomic coding DNA from a diverse sample of organisms obtained from different taxa in the CUTG database. The second dataset includes measurements from 16 chemical sensors exposed to six different gases at various concentration levels. The third dataset involves smartphone-based recognition of human activities and postural transitions, performing various activities. All these tasks represent classification problems, and we evaluate performance based on the error of each algorithm.

First, we assessed the effectiveness of our NMs and other methods across various node complexities. This evaluation allows us to understand how the NMs perform with different levels of complexity. The nodes were initially organized using NN, DEQ, DAG as well as OPTNET, and their percentage of error was observed. Our experiments involved comparing the performance of NMs, examining their performance with frequently-used $L1$ regularization and the proposed $NM$ regularization, as introduced earlier. The results, as presented in the tables, reveal that our novel structure consistently achieves superior results in most cases. Additionally, the performance of $NM$ regularization surpasses that of the benchmark, $L1$ regularization in most situations. With the optimal performance achieved with a suitable number of nodes, our $NM$ regularization is shown to perform the best across all the datasets.

For all datasets and node variations within the graph, NN consistently exhibits improved results when nodes are organized into neural modules. The efficacy of our NMs can be further enhanced through regularization, as previously explained. Notably, even without regulation, our NMs outper-

Table 1: The error of algorithms for Codon Usage Dataset

|       | 40 Nodes | 60 Nodes | 80 Nodes | 100 Nodes | 200 Nodes | 300 Nodes |
|-------|----------|----------|----------|-----------|-----------|-----------|
| NN    | 0.2128   | 0.1917   | 0.1854   | 0.1964    | 0.1702    | 0.1839    |
| DEQ   | 0.1714   | 0.1839   | 0.2136   | 0.1980    | 0.1714    | 0.2911    |
| DAG   | 0.2410   | 0.2152   | 0.2027   | 0.2050    | 0.2074    | 0.1987    |
| OPTNET| 0.1792   | 0.1706   | 0.2152   | 0.2034    | 0.1745    | 0.1901    |
| NMs   | 0.1761   | 0.1557   | 0.1643   | 0.1604    | 0.1792    | 0.1432    |
| NMs&L1| 0.1753   | 0.1549   | 0.1964   | 0.1591    | 0.1776    | 0.1549    |
| NMs&NM| **0.1495** | **0.1510** | **0.1505** | **0.1549** | **0.1659** | **0.1408** |

Table 2: The error of algorithms for Gases Concentration Dataset

|       | 40 Nodes | 60 Nodes | 80 Nodes | 100 Nodes | 200 Nodes | 300 Nodes |
|-------|----------|----------|----------|-----------|-----------|-----------|
| NN    | 0.1669   | 0.1293   | 0.1113   | 0.1414    | 0.1639    | 0.2391    |
| DEQ   | 0.1188   | 0.1233   | 0.1323   | 0.1098    | 0.1278    | 0.1474    |
| DAG   | 0.2677   | 0.1835   | 0.1248   | 0.1263    | 0.2857    | **0.1293** |
| OPTNET| 0.1308   | 0.1774   | 0.1248   | 0.1549    | **0.1113** | 0.2541    |
| NMs   | 0.1023   | 0.1098   | 0602     | 0.0767    | 0.1714    | 0.2135    |
| NMs&L1| 0.0932   | 0.0992   | 0.0586   | 0.0602    | 0.1714    | 0.2075    |
| NMs&NM| **0.0752** | **0.0962** | **0.0511** | **0.0301** | 0.1684    | 0.1970    |

form the traditional structure. After implementing $NM$ regularization, our approach demonstrates enhanced performance.

## 4.2 Efficiency of Neural Modules

In this section, we delve into the optimization of Neural Module efficiency through the application of $NM$ regularization. As previously mentioned, every structure considered is a subgraph of a fully connected graph, with the initial general graph acting as the search space for our model. Our $NM$ regularization serves as a potent mechanism for structural optimization, enhancing the effectiveness and balance of neural modules. The performance of these modules, particularly for nodes numbering less than 300, was demonstrated in the preceding section, where it was shown to yield superior outcomes.

To assess the efficiency of our $NM$ regularization, we present a comparison of the model's running time with DEQ across varying complexities, represented by node counts below 300. Figure 4, Image A, illustrates that the efficiency of NM regularization significantly outperforms DEQ, especially with a larger number of nodes. This superiority is attributed to $NM$ regularization's ability to create multiple independent neural modules, which efficiently mitigate computational complexity.

For networks with a larger number of nodes, we can leverage the parallel processing capabilities of Neural Modules to enhance the efficiency of our framework, as previously discussed. $NM$ regularization facilitates the creation of multiple independent and well-balanced neural modules, which are inherently suited for parallel computing, particularly when transitioning to GPU-based computation.

Table 3: The error of algorithms for Postural Transitions Dataset

|       | 40 Nodes | 60 Nodes | 80 Nodes | 100 Nodes | 200 Nodes | 300 Nodes |
|-------|----------|----------|----------|-----------|-----------|-----------|
| NN    | 0.1062   | 0.1099   | 0.1008   | 0.1069    | 0.0811    | 0.1245    |
| DEQ   | 0.1232   | 0.2755   | 0.0882   | 0.0889    | 0.0859    | 0.0658    |
| DAG   | 0.1154   | 0.1130   | 0.1184   | 0.0937    | 0.1025    | 0.1639    |
| OPTNET| 0.1188   | **0.1055** | 0.0940   | 0.1164    | 0.0994    | 0.1639    |
| NMs   | 0.1018   | 0.2559   | 0.0886   | 0.0699    | 0.1035    | 0.1059    |
| NMs&L1| 0.0865   | 0.2538   | 0.0821   | 0.0665    | 0.0787    | 0.0726    |
| NML&NM| **0.0709** | 0.2416   | **0.0724** | **0.0570** | **0.0631** | **0.0635** |

In this extension, we increase the node count from 300 to 3000 and incorporate GPU acceleration to compute the algorithms more efficiently.

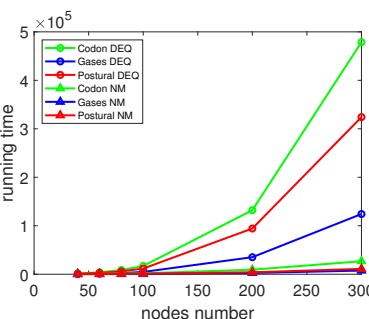 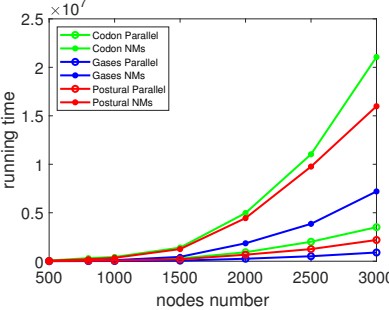

(a) Nodes' number up to 300. Neural modules vs.DEQ.

(b) Nodes' number up to 3000. Neural modules parallel vs. Neural modules.

Figure 4: Neural Modules offer significant advantages over DEQ in terms of computational efficiency. Moreover, through parallel computation, $NM$ regularization provides an opportunity to extend the applicability of this framework to larger-scale applications. By harnessing the power of parallel processing, $NM$ regularization allows for the efficient handling of increased complexity, making it a promising approach for tackling more extensive computational challenges.

In Figure 4, Image B, we conducted a comparison between the running times of Neural Modules operating in parallel and Neural Modules without parallelization on large realistic models. For these experiments, we utilized 12 threads. Our results indicate that the parallel implementation of $NM$ regularization offers a computational speedup of approximately 6 to 7 times. This demonstrates the substantial efficiency gains achievable through parallel processing in the context of NM regularization.

Furthermore, for networks with a higher number of nodes, parallel $NM$ regularization also yields increasingly better results, as demonstrated in Figure 5. In our analysis, we compared methods such as DEQ, DAG, and OPTNET, and observed that when the number of nodes exceeds 300, these methods generally result in overtime. We established the average performance of these three models with node counts within 300 as our baseline for comparison.

Additionally, to assess the performance of our framework on applications with larger node numbers, we included the NN structure as a reference in our experiments. We discovered that our framework performs better on larger models. This indicates that our framework is more efficient and accurate, particularly when dealing with complex, large-scale neural network structures.

### 4.3 THE EFFECT OF $NM$ REGULARIZATION

At last, Figure 6 presents examples of neural modules generated by our $NM$ regularization, where each black square denotes a neural module. The image on the left shows the neural modules generated at the first iteration, where the effects of $NM$ regulation are not yet apparent. The image on the right, at the 10000th iteration, illustrates how $NM$ regulation achieves independence and balance within the generated neural modules. $NM$ regularization helps break large neural modules into smaller, more manageable neural modules. This segmentation leads to improved performance and efficiency, as previously introduced.

## 5 CONCLUSION

This study introduces a novel general graph structure designed for the NNs, aiming to improve performance by facilitating significant information transfer. We address structural bias analysis for the current tree-like structure. Our model employs a synchronization method for the simultaneous calculation of node values, thereby fostering collaboration within neural modules. Additionally, we propose a novel $NM$ regularization method that encourages the learned structure to prioritize critical connections and automatically formulate multiple independent, balanced neural structures, which

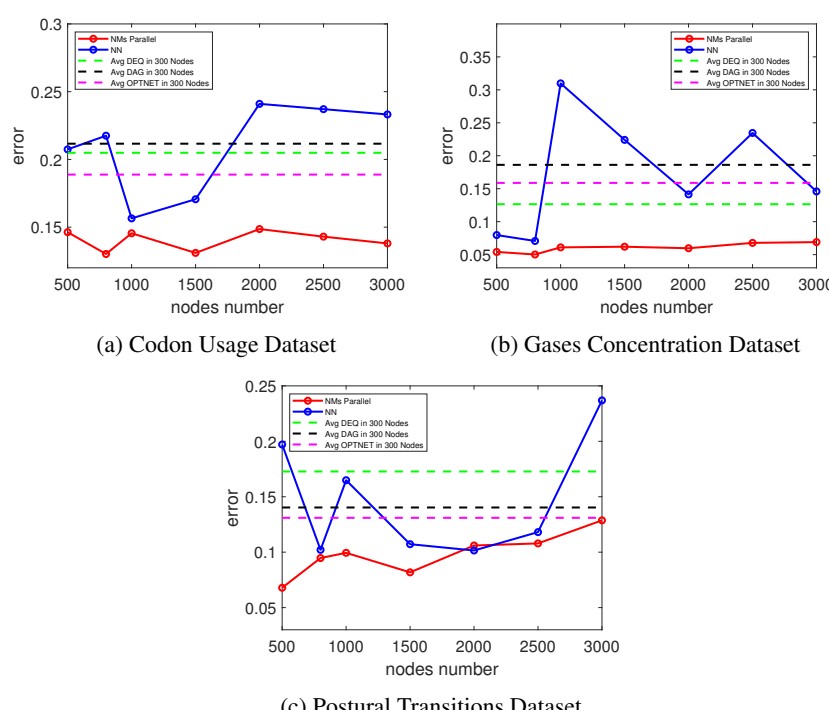

(a) Codon Usage Dataset        (b) Gases Concentration Dataset

(c) Postural Transitions Dataset

Figure 5: The performance on the 3 Datasets with larger nodes' number.

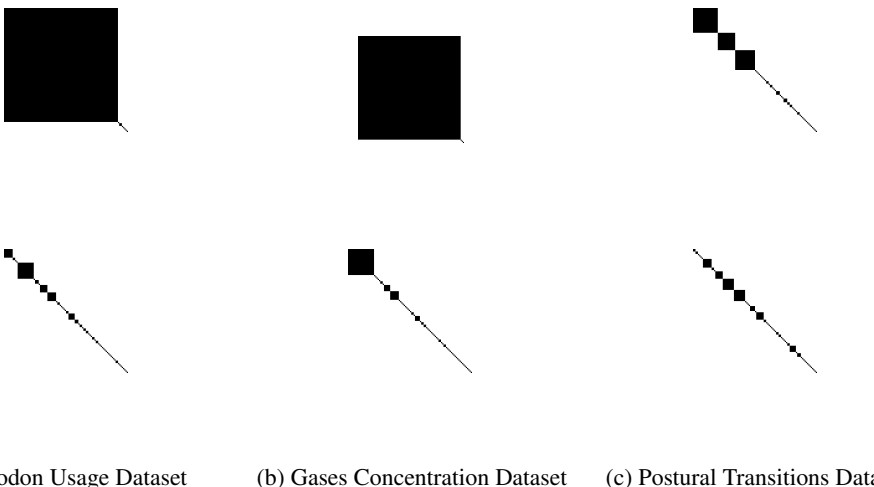

(a) Codon Usage Dataset      (b) Gases Concentration Dataset      (c) Postural Transitions Dataset

Figure 6: Neural modules generated by our $NM$ regularization for the Codon Usage Dataset, the Gases Concentration Dataset, and the Postural Transitions Transitions Dataset with each black square denoting a neural module. The left image depicts the neural modules at the first iteration, and the right image shows the neural modules at the 10000th iteration.$NM$ regularization helps break large neural modules into smaller, more manageable neural modules.

would help to achieve better efficiency by calculation in parallel. This approach not only reduces the computational load associated with managing more nodes but also improves the performance by mitigating overfitting. Quantitative experimental results affirm the superiority of our proposed method over traditional structures for NNs.

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

## A   THE SIGNIFICANCE OF OUR STUDY

Particularly, what sets our research apart is its inspiration drawn from the intricate dynamics of biological neural systems with billions of neural units. Many units are influenced by each other and generate a collaborative effect. Unlike the traditional stacked unit approach, our approach mirrors the cooperative nature of biological neural modules. In these systems, multiple neural units work together to perform precise functional tasks, resulting in exquisite performance as illustrated in the 2nd image of Figure 8. Our innovation seeks to bridge the gap between artificial and biological neural networks, thus propelling NN structures toward the performance observed in their natural counterparts.

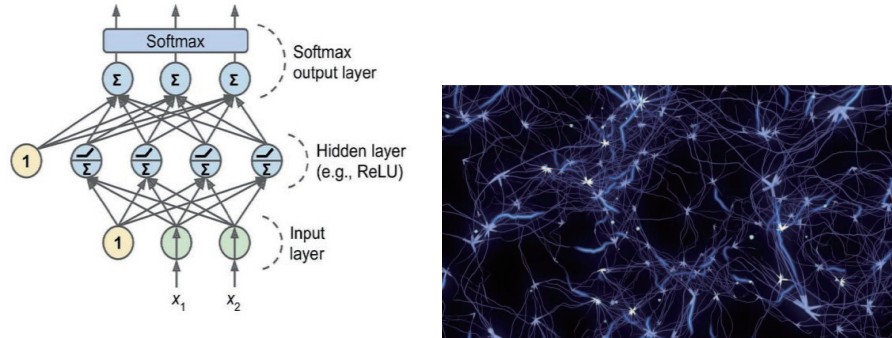

Figure 7: The left image illustrates the typical structure of an NN, while the right image depicts a biological neural network, showcasing its inherently more complex connectivity patterns.

The NNs structure in the current framework fails to incorporate a cyclic graph which indicates a synchronous structure that makes the collaboration of the neural units in NNs. The current tree-like structure is asynchronous, the cyclic graph would produce an infinite loop, as illustrated in Figure 9. Our framework facilitates collaborative interactions among neural units. In this paper, we present a method that enables synchronous communication among nodes within the neural module. This transformative adjustment enhances information processing and increases the overall capacity of NN structures.

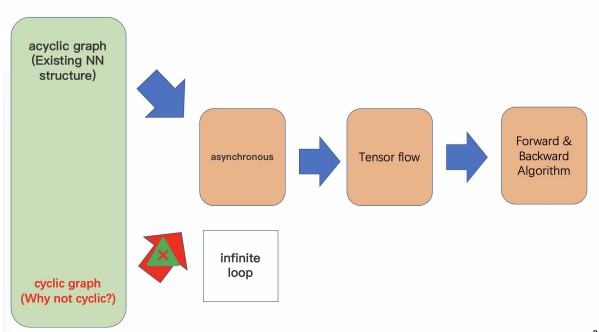

Figure 8: Illustration of the limitations preventing the application of a cyclic graph which is a synchronous structure within the current tree-like NN structure.

## B  THE CONNECTION WITH NNs

In this section, we argue that the existing tree-like NN structure in fact is a special case of our framework. The comparison is detailed in the following table, the asynchronous process for the current tree-like structure can also be formalized by our system of equations for our framework.

Table 4: table B

| | NNs | NMs |
|---|---|---|
| Model | $\mathcal{T} = \{N^0, E^1, N^1, ..., E^m, N^m\}$ | $\mathcal{NM} = \{N^0, E^1.\mathcal{G}, E^m, N^m\} = \{N^0, E^1.E, N, E^m, N^m\}$ |
| Variables | $\mathcal{T} = \begin{pmatrix} \mathcal{X}^0 = [1, X^0] \\ \mathcal{W}^0 = [bias_1, W^1] \\ \mathcal{X}^1 = [1, X^1] \\ ... \\ \mathcal{W}^m = [bias_m.W^m] \\ X^m \end{pmatrix}$ | $NM = \begin{pmatrix} X^0 \\ W^1 \\ W \\ X \\ W^m \\ X^m \end{pmatrix}$ |
| FP | $\begin{cases} \mathcal{X}^0 \cdot \mathcal{W}^{1T} = X^1 \\ \mathcal{X}^1 \cdot \mathcal{W}^{2T} = X^2 \\ ... \\ \mathcal{X}^{m-1} \cdot \mathcal{W}^{mT} = X^m \end{cases}$ | $\begin{cases} w_{1,1} + \sum\limits_{j \neq 1} f(x_j) \cdot w_{j,1} + \sum\limits_{j=1}^{|N^0|} x_j^0 \cdot w_{j,1}^1 = x_1 \\ w_{2,2} + \sum\limits_{j \neq 2} f(x_j) \cdot w_{j,2} + \sum\limits_{j=1}^{|N^0|} x_j^0 \cdot w_{j,2}^1 = x_2 \\ ... \\ w_{p,p} + \sum\limits_{j \neq p} f(x_j) \cdot w_{j,p} + \sum\limits_{j=1}^{|N^0|} x_j^0 \cdot w_{j,p}^1 = x_p \\ X \cdot W^{mT} = X^m \end{cases}$ |
| BP | $\begin{cases} \nabla X^m \cdot g'(X^m) \cdot W^m = \nabla X^{m-1} \\ ... \\ \nabla X^2 \cdot f'(X^2) \cdot W^2 = \nabla X^1 \end{cases}$ | $\begin{cases} \sum\limits_{j \neq 1} \nabla x_j \cdot f'(x_j) \cdot w_{1,j} + \sum\limits_{j=1}^{|N^m|} \nabla x_j^m \cdot g'(x_j^m) \cdot w_{1,j}^m = \nabla x_1 \\ \sum\limits_{j \neq 2} \nabla x_j \cdot f'(x_j) \cdot w_{2,j} + \sum\limits_{j=1}^{|N^m|} \nabla x_j^m \cdot g'(x_j^m) \cdot w_{2,j}^m = \nabla x_2 \\ ... \\ \sum\limits_{j \neq m} \nabla x_j \cdot f'(x_j) \cdot w_{p,j} + \sum\limits_{j=1}^{|N^m|} \nabla x_j^m \cdot g'(x_j^m) \cdot w_{p,j}^m = \nabla x_p \end{cases}$ |

From the table, we can see that the traditional NN structure represents a special case of our framework where the system of equations is solved asynchronously. In the tree-like structure, the coefficient matrix of the equations is composed of the parameter matrices for each level, and these coefficients are positioned near the diagonal. In contrast, our framework generalizes the coefficient to form a full matrix across the 1 to the $p-1$ level, indicating a transition from a tree-based structure to a general graph structure.

## C  THE MOTIVATION OF REORGANIZING THE NEURAL NETWORKS FROM TREE-LIKE STRUCTURE TO GENERAL GRAPH STRUCTURE

For the traditional tree-like structure, the asynchronous forward and backward process can also be considered to solve a system of equations as shown in Table 4. In our framework, the feed-forward processes for traditional tree-like structures are essential to solve the equations as shown in Figure 9, which is similar to the backward process.

For traditional tree-like structures, the asynchronous forward and backward processes can indeed be utilized to address the solution of a system of equations, as illustrated in Table 4. In our framework, the feed-forward processes for traditional tree-like structures are essential to solve the equations as shown in Figure 9, This process bears resemblance to the backward propagation phase.

Assign the input and bias-related values to the right side. Allocate the values associated with each node to the left side. Consequently, the equations are reformulated as shown in Figure 10.

Then, the coefficient $\mathcal{C}$ for the tree-like structure in our framework would be as Image A in Figure 11.

The tree-like structure in our framework is indeed more constrained for each node within the structure. According to the Universal Approximation Theorem, the flexibility of a Neural Network (NN) is contingent upon the number of neurons within a single hidden layer. This implies that for any nodes in a tree structure if they are solely dependent on the neurons in the preceding layer, their

$$\begin{cases} w_1^1 + \sum_{j=1}^{|N^0|} x_j^0 \cdot w_{j,1}^1 = x_1^1 \\[2mm] w_2^1 + \sum_{j=1}^{|N^0|} x_j^0 \cdot w_{j,2}^1 = x_2^1 \\[2mm] ... \\[2mm] w_{|N^1|}^1 + \sum_{j=1}^{|N^0|} x_j^0 \cdot w_{j,|N^1|}^1 = x_{|N^1|}^1 \\[2mm] ...... \\[2mm] w_1^{m-1} + \sum_{j=1}^{|N^{m-2}|} f(x_j^{m-2}) \cdot w_{j,1}^{m-1} = x_1^{m-1} \\[2mm] w_2^{m-1} + \sum_{j=1}^{|N^{m-2}|} f(x_j^{m-2}) \cdot w_{j,2}^{m-1} = x_2^{m-1} \\[2mm] ... \\[2mm] w_{|N^{m-1}|}^{m-1} + \sum_{j=1}^{|N^{m-2}|} f(x_j^{m-2}) \cdot w_{j,|N^{m-1}|}^{m-1} = x_{|N^{m-1}|}^{m-1} \end{cases}$$

Figure 9: Tadition tree-like structure for NN is essentially to solve these equations in our framework.

$$\begin{cases} -x_1^1 = -w_1^1 - \sum_{j=1}^{|N^0|} x_j^0 \cdot w_{j,1}^1 \\[2mm] -x_2^1 = -w_2^1 - \sum_{j=1}^{|N^0|} x_j^0 \cdot w_{j,2}^1 \\[2mm] ... \\[2mm] -x_{|N^1|}^1 = -w_{|N^1|}^1 - \sum_{j=1}^{|N^0|} x_j^0 \cdot w_{j,|N^1|}^1 \\[2mm] ...... \\[2mm] \sum_{j=1}^{|N^{m-2}|} f(x_j^{m-2}) \cdot w_{j,1}^{m-1} - x_1^{m-1} = -w_1^{m-1} \\[2mm] \sum_{j=1}^{|N^{m-2}|} f(x_j^{m-2}) \cdot w_{j,2}^{m-1} - x_2^{m-1} = -w_2^{m-1} \\[2mm] ... \\[2mm] \sum_{j=1}^{|N^{m-2}|} f(x_j^{m-2}) \cdot w_{j,|N^{m-1}|}^{m-1} - x_{|N^{m-1}|}^{m-1} = w_{|N^{m-1}|}^{m-1} \end{cases}$$

Figure 10: Assign the values associated with inputs and bias to the right-hand side. Place the values related to each node on the left-hand side. This arrangement allows us to extract the coefficient $\mathcal{C}$ for our framework.

capacity for complex function approximation is severely limited. Typically, nodes in the first layer are represented as a linear transformation of the input values.

Some existing work has attempted to generalize the structure to a Directed Acyclic Graph (DAG), such as in the case of ResNet. In these architectures, additional weights are introduced to the Lower Triangular of the coefficient matrix $\mathcal{C}$. Let these weights be represented by $V$. Consequently, the matrix $\mathcal{C}$ is depicted in Image B of Figure 11. This modification allows for a more flexible and expressive model, which can better approximate complex functions and handle larger datasets, aligning with the theorem's assertion that a single hidden layer with a sufficient number of neurons can approximate any continuous function on a compact subset of the real numbers.

$$
\begin{pmatrix}
-E^1 & 0 & 0 & 0 & ... & 0 & 0 \\
W^1 & -E^2 & 0 & 0 & ... & 0 & 0 \\
0 & W^2 & -E^3 & 0 & ... & 0 & 0 \\
0 & 0 & W^3 & -E^4 & ... & 0 & 0 \\
...... & & & & & & \\
0 & 0 & 0 & 0 & ... & W^{m-2} & E^{m-1}
\end{pmatrix}
$$

(a) The Coefficient $\mathcal{C}$ for the Tree-like Structure and $E$ is the identity matrix for each level.

$$
\begin{pmatrix}
-E^1 & 0 & 0 & 0 & ... & 0 & 0 \\
W^1 & -E^2 & 0 & 0 & ... & 0 & 0 \\
V^1 & W^2 & -E^3 & 0 & ... & 0 & 0 \\
V^1 & V^2 & W^3 & -E^4 & ... & 0 & 0 \\
...... & & & & & & \\
V^1 & V^2 & V^3 & V^4 & ... & W^{m-2} & E^{m-1}
\end{pmatrix}
$$

(b) The Coefficient $\mathcal{C}$ for the DAG or ResNet Structure and $E$ is the identity matrix for each level.

Figure 11: The Coefficient $\mathcal{C}$ for traditional NN Structure

The Lower Triangular Coefficient matrix indeed reaches the limits of asynchronous structures. However, the potential of the Upper Triangular matrix remains largely untapped. In our work, we extend the asynchronous structure to a synchronous one, breaking through the limitations of the Lower Triangular and generalizing the coefficient matrix $\mathcal{C}$. Within our Neural Module framework, the $\mathcal{C}$ is depicted in Figure 12, showcasing a more comprehensive and interconnected structure that allows for greater flexibility and performance.

$$
\begin{pmatrix}
-1 & w_{2,1} & w_{3,1} & w_{4,1} & ... & w_{p-1,1} & w_{p,1} \\
w_{1,2} & -1 & w_{3,2} & w_{4,2} & ... & w_{p-1,2} & w_{p,2} \\
w_{1,3} & w_{2,3} & -1 & w_{4,3} & ... & w_{p-1,3} & w_{p,3} \\
w_{1,4} & w_{2,4} & w_{3,4} & -1 & ... & w_{p-1,4} & w_{p,4} \\
...... & & & & & & \\
w_{1,p} & w_{2,p} & w_{3,p} & w_{4,p} & ... & w_{p-1,p} & -1
\end{pmatrix}
$$

Figure 12: The Coefficient $\mathcal{C}$ for our NM Structure

In our framework, we have enhanced the representational capacity of each neuron, thereby unlocking the full potential of Neural Networks (NNs). Concurrently, we have eliminated the structural bias that is typically inherent in predefined structures such as traditional tree structures or Directed Acyclic Graphs (DAGs). This innovation allows our framework to be more adaptable and less constrained by the limitations of fixed architectural biases, leading to a more flexible and effective NN design.

For large coefficient matrices $\mathcal{C}$, solving the system of equations can indeed be challenging. Our framework addresses this by proposing $NM$ regularization, which serves as an approximation method for $\mathcal{C}$ and incorporates parallel computation to enhance efficiency. However, it's important to note that $NM$ regularization is an approximation technique, and it operates on the complete set of the coefficient matrix $\mathcal{C}$ as previously introduced. This approach allows us to break down the complex system into more manageable parts, which can then be solved in parallel, significantly reducing the computational burden and improving the overall performance of the framework.

## D  THE CONNECTION WITH DEQ

Previous research has observed the existence of a stable point in an infinite-level NN structure with identical weights. DEQ addresses this issue by modeling the problem across infinite levels and

simplifying it to an implicit function. Specifically, achieving a fixed point is equivalent to solving the root of the implicit function. In our approach, we find that our general graph structure can also be managed by solving a system of functions.

In fact, DEQ focuses on infinite levels with the same weight. Suppose treating the infinite path as a circle. The essential research object of DEQ is also a cyclic graph. In this paper, we unveil the fundamental nature of the fixed point, recognizing its role as a solution within our synchronous neural module structure. Our neural module not only assists in finding its essence but also enables concurrent visualization of the implicit function as shown in Figure 10 and Figure 11.

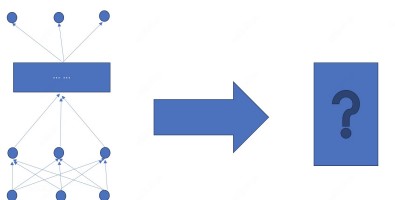

Figure 13: For the infinite structure, DEQ abstracts an implicit function to solve the fixed point.

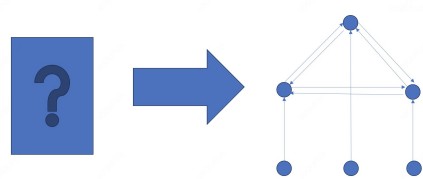

Figure 14: Our model tries to find the essence of the implicit function, a general graph structure.

Furthermore, adhering to the Universality of Single-layer DEQ, it is established that multiple implicitly hidden layers are tantamount to a single implicitly hidden layer. As a result, DEQ encounters limitations in managing multiple implicitly hidden layers.

Most importantly, DEQ struggles when managing a larger number of nodes as introduced in the paper. These layers must solve the implicit function. In our paper, by considering implicit hidden layers from infinite levels to elegant general graphs, we would organize the neural units well to optimize the efficiency as well as the performance of the model. To address this, we introduce $NM$ regularization that organizes these nodes into neural modules. In contrast to the DEQ, our framework provides a methodology for handling multiple neural modules. Our analysis of the regularization encompasses both theoretical foundations and experimental parameters, enabling our framework to manage larger graphs effectively. Furthermore, the organized independent, balanced neural models achieve superior results on both efficiency and performance, as previously demonstrated. These neural modules improve performance by reducing overfitting and improving efficiency through parallel computation. Previous work like DEQ did not discuss how to organize the neural units to optimize efficiency and performance.

## E  THE CONNECTION WITH OPTNET

In earlier research, OPTNET examines the interconnections among nodes within the same level in traditional tree-like structures. Their approach involves implementing Quadratic Programming within these nodes, which introduces significant bias. Furthermore, OptNet uses parameters derived from quadratic problems for backpropagation, determined by nodes from the previous layer which brings difficulties to regularization as illustrated before. The defects of OPTNET make it challenging to fine-tune its performance as well as overall efficiency.

In contrast, our neural module is nonlinear and exhibits sufficient flexibility to accommodate any compact function without introducing bias, as affirmed by the Universal Neural Module Approxi-

Table 5: Experimental Environment

| CPU | Gen Intel(R) Core(TM) i9-12900H 2.90 GHz |
|---|---|
| Cores | 12 |
| Memory | 32G |
| GPU | NVidia GeForce RTX 3060 |
| Graphics Memory | 12G |
| Norm Function | L1 |
| Other Parameters | Refer to Figure 15 |

mation Theorem. Additionally, we can readily control the complexity of the parameters to enhance performance and efficiency simultaneously.

## F THE SUPPLEMENTARY MATERIALS FOR THE EXPERIMENTS

In this section, we provide a detailed description of the experimental setup, which is presented in Table 5. Additionally, as evidenced in this section, the parameters within our framework are adjustable, allowing for optimization and fine-tuning to achieve the best performance.

In our experimental setup, for the regularization function $norm$, we employ the absolute value of the weights, akin to $L1$ regularization. This choice is strategic, as it helps to expand the neural module while also highlighting a few critical edges during each iteration. In the context of larger neural modules, these critical edges may become less frequent as the process of $NM$ regularization unfolds. This approach encourages the model to concentrate on the most significant connections, thereby improving the efficiency and performance of the neural module.

In real applications, other regular functions can be tried for $norm$ to get better performance.

### F.1 THE OPTIMIZATION OF THE NEURAL MODULES

In this section, we explore the process of optimizing the structure of Neural Modules using $NM$ regularization. As discussed earlier, the results obtained have shown that $NM$ regularization would yield better results. Here, we investigate the parameter-setting strategies for our $NM$ regularization.

The images in Figure 15 illustrate the impact of different parameters on the error rates of our $NM$ regulation across various percentages of approximations(controlled by parameter $\gamma$) for the Codon Usage Dataset. We found that the performance can be optimized by fine-tuning the $NM$ regularization and the percentages of approximations(controlled by parameter $\gamma$). Higher approximation percentages and regularization parameters produce more Neural Modules, which help reduce overfitting problems and in turn, enhance performance as well as robustness. However, overly sparse NMs could introduce extra bias and degrade performance. After standardization, the $NM$ regularization parameter with 100 nodes performs best at 0.2 and the approximation on top 0.05 edges by controlling parameter $\gamma$ for the Codon Usage Dataset. However, more stable performance appears on the approximation of top 0.03 edges.

In Figure 15, we also offer this on the Gases Concentration Dataset and the Postural Transitions Dataset. We reached a similar conclusion that performance can be well optimized by tuning both the $NM$ regularization and the parameters of approximation. We observe that for these two datasets, the higher parameter $\gamma$ and $NM$ regularization parameters result in better performance. $NM$ regularization helps break large neural modules into smaller, more manageable neural modules. This segmentation leads to improved performance as previously introduced. However. compared with the Codeon Usage Data, the approximation can not to too large. On average, for 100 nodes, the $NM$ regularization parameter usually performs best at 0.2 with the approximation on top 0.01 edges by controlling parameter $\gamma$.

### F.2 TRADE OFF PERFORMANCE AGAINST EFFICIENT ON TRADITIONAL NN

In this section, we propose an additional experiment to compare our framework with widely used traditional Neural Networks(NNs). The aim of this experiment is to assess how well our method

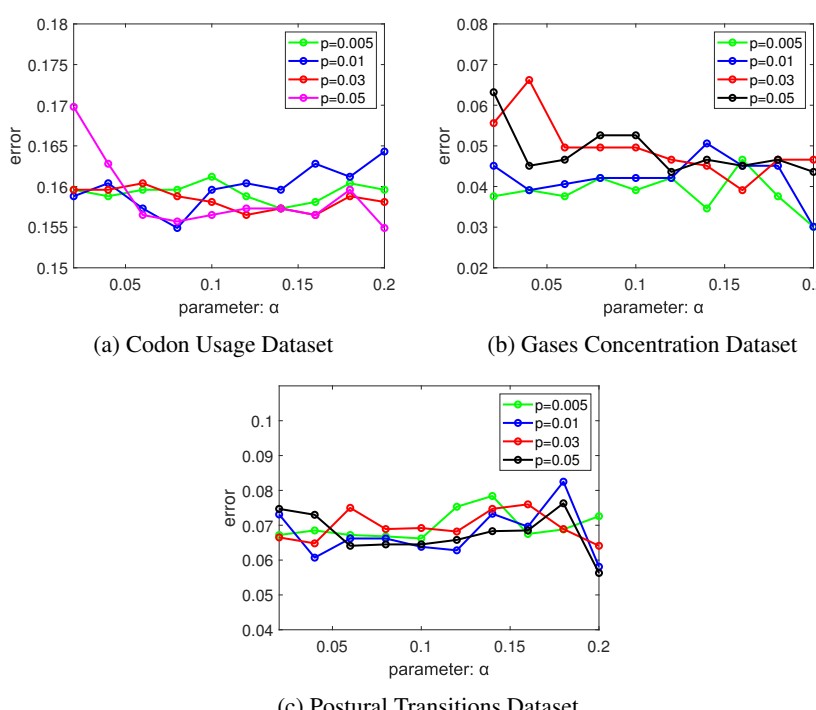

(a) Codon Usage Dataset

(b) Gases Concentration Dataset

(c) Postural Transitions Dataset

Figure 15: Parameter setting for $NM$ regularization.

performs in comparison to traditional NNs, especially when these networks have an adequate number of nodes, and to evaluate the trade-offs regarding efficiency. To prevent overfitting, we employ a more complex dataset with thousands of features. We utilize a web graph, a page-page graph of verified Facebook sites. In this graph, nodes represent official Facebook pages, and the edges represent mutual likes between these sites. This dataset can also be found in the UCI Machine Learning Repository.

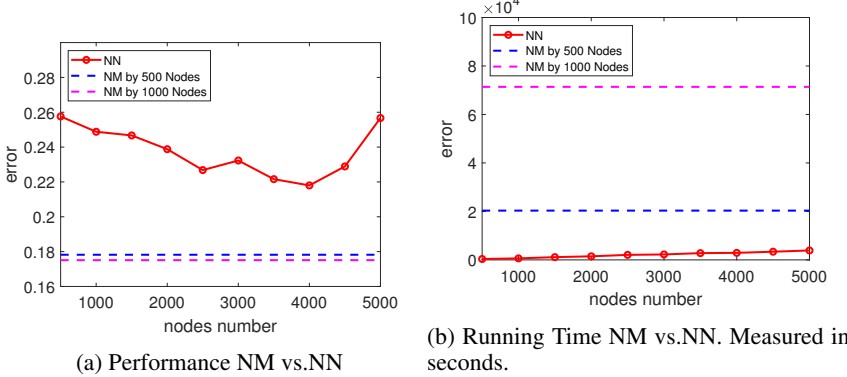

(a) Performance NM vs.NN

(b) Running Time NM vs.NN. Measured in seconds.

Figure 16: NM vs. NN

From Image A Figure 16, we observed that the performance of traditional Neural Networks(NNs) deteriorates at 4000 nodes due to overfitting, which is also the best result it achieves. In contrast, our NM framework yields a more than 40 percent improvement in accuracy at 500 nodes and a even better result at 1000 nodes. On the other hand, as depicted in Figure B Figure 16, the improvement is with the count of several hours on efficiency. However, both performance and running time can be controlled by parameters such as sparsity. This indicates that our method is well-suited for tasks

that prioritize accuracy. Users can weigh the trade-off between employing our method to enhance accuracy and the associated efficiency costs.

### F.3 THE EFFECT OF $NM$ REGULARIZATION

From the experiments, $NM$ regularization can better capture the trend of the weights in each iteration. It enables accurate weight regularization and formulates effectively balanced neural modules. More importantly, these independent, balanced neural modules bring additional benefits for efficiency. As previously mentioned, the overall efficiency is determined by the size of the largest neural module, especially when they are processed in parallel.

To analyze the effect of $NM$ regularization, consider the probability of the edge's weight being lower than $\gamma$ denoted as $\theta$ in the current iteration. For a neural module with $q$ nodes, the probability of a new edge integrating into the neural module would be denoted as

$$1 - \theta^q \tag{11}$$

. In the case of a small $q$ value, $NM$ regularization seeks to increase the probability. Conversely, for larger $q$ values, NM regularization works to decrease the probability. Thus, NM regularization endeavors to minimize the disparity in the probability of a new edge integrating into the neural modules with varying sizes. This effect is illustrated in the first image of Figure 4.

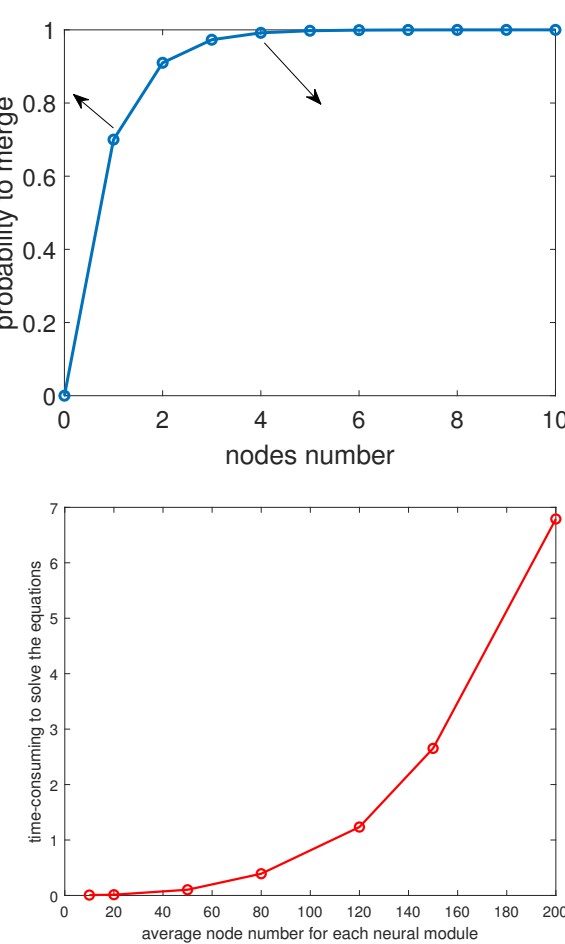

Figure 17: Image A shows the analysis of $NM$ Regularization and image B displays the time-consumption for different sizes of neural modules which indicates parallel computing's advantage.

Image A in Figure 17 illustrates the analysis of $NM$ Regularization and how balanced Neural Modules are formed through this process. Image B in Figure 17 shows the time consumption associated

with different sizes of neural modules. As previously mentioned, the neural modules generated are independent and can be calculated in parallel. For instance, if a neural module comprises 50 nodes, the time to solve the system of equations required would be reduced to just 10ms in our experiments. Thus, $NM$ regularization enables sufficient management of larger node counts.

## G    NEURAL MODULE IS ONLY FOR APPROXIMATION TO SOLVE THE SYSTEM OF EQUATIONS

In the forward and backward process, the solution involves a system of nonlinear equations. The presence of numerous nodes in the neural module can lead to performance issues. To address this challenge, we propose $NM$ regularization, which organizes multiple independent, balanced neural modules. Breaking down a large system of nonlinear equations into more manageable systems significantly reduces the computational burden. Furthermore, solving these equations in parallel would further enhance the overall efficiency.

Note that, we update all the edges in every iteration, which provides additional benefits such as enhanced robustness. For the absolute value of the edge's weight lower than $\gamma$, it also chance to grow larger than $\gamma$ in the coming iteration. That means every edge in the graph has the opportunity to change, thereby retaining the dynamism of Neural Modules and exerting our model's performance.

## H    THE PROOF OF THE UNIVERSAL NEURAL MODULE APPROXIMATION THEOREM

Note that the array of equations involves an activation function. Therefore, in our model, the system of functions is non-linear and possesses sufficient flexibility to fit any appropriate function, according to our Universal Neural Module Approximation Theorem.

*Proof.* In the forward process, the system of equations constructs implicit functions across all nodes within the neural module. Subsequently, any node $n_i$ can be transformed into an explicit function. According to the Universal Approximation Theorem (Cybenko, 1989) (Hornik & White, 1990) (Leshno & Schocken, 1993a), which states that for any continuous function on a compact set, there exists a one-hidden-layer feed-forward network capable of approximating the function, our system satisfies this condition. This is because our array of equations involves an activation function, fulfilling the requirement of the theorem. Therefore, the proof is complete.    □

