# OpenReview forum: "Automatic Organization of Neural Modules for Enhanced Collaboration in Neural Networks"
_ICLR.cc/2025/Conference — Submitted to ICLR 2025_

### Official Review · Reviewer_589V · 2024-11-03

**Soundness:** 2
**Presentation:** 2
**Contribution:** 2
**Rating:** 3
**Confidence:** 4

**Summary:**

This manuscript proposes to organize the neurons in a neural network to a graph-based structure along with an regularization method. While the experimental results demonstrate advancement, the motivation of reorganizing the neural networks from tree-like structure to flat graph structure requires further clarification. The contribution of this work is not convincing since existing works such as reservoir computing have also adopted such structure.

**Strengths:**

1. The rethinking and redesigning of the neural network structure is in urgent need and the idea of this work originates from real biological neurons.
2. The methods and contends in this paper are easy to follow.
3. Experimental results demonstrate advancement of the proposed framework.

**Weaknesses:**

1. The motivation of reorganizing the neural networks from tree-like structure to flat graph structure requires further clarification.
2. The literature review to existing works is not extensive, while the differences with similar works such as reservoir computing have not been explained.
3. The figures are not expressive.
4. While the parameter \gamma in definition 1 lies within the core of this work, its usage and selection are not detailedly described, which should be explicitly explained in equations or pseudocode.
5. The proposed NM regularization method requires theoretical verification. While the authors claimed that "a larger value of z would cause the edges connected to these subgraphs to tend toward zero", how to initialize z is not described.
6. The experimental results require detailed description to the experimental environment as in other referenced works.
7. The results in section 4.2-4.4 are quite foreseeable, which bring minor information to the readers.
8. The contents in appendix are redundant and also bring little information.

**Questions:**

See Weaknesses.

---

> ### Author Response · Authors · 2024-11-24
>
> Dear Respected Reviewer：
>
> Thank you for your insightful feedback. I have carefully considered your comments and have made the following revisions to address your concerns. We have revised the paper and incorporated the new code with parallel computing capabilities in the supplementary material for our experiments.  The code is for our experiments on the Postural Transitions dataset.
>
> 1.The motivation of reorganizing the neural networks from tree-like structure to flat graph structure requires further clarification.
>
> Answer: You requested further clarification on the rationale for transitioning from a tree-like to a graph structure in neural networks. This new theoretical explanation is now detailed in Section C of the appendix in our updated submission.
>
> 2.The literature review to existing works is not extensive, while the differences with similar works such as reservoir computing have not been explained.
>
> Answer: We acknowledge the need for a more comprehensive literature review and a clearer distinction from similar works like reservoir computing. These additions are now included in Section 2 of our revised paper. Reservoir computing, which is a type of recurrent neural network with randomly connected neurons and fixed weights, is contrasted with our Neural Module appThe figures are not expressive.roach, which is based on a balanced system where both weights and structure are adaptable， offering enhanced capabilities for our framework.
>
> 3.The figures are not expressive.
>
> Answer：We have revised the figures and added new ones to provide more insightful and valuable information, as detailed in our updated submission.
>
> 4.While the parameter \gamma in definition 1 lies within the core of this work, its usage and selection are not detailedly described, which should be explicitly explained in equations or pseudocode.
>
> Answer: The role and selection criteria for the parameter γ,  have been explicitly detailed in Equation 1 of the revised paper.
> 5.The proposed NM regularization method requires theoretical verification. While the authors claimed that "a larger value of z would cause the edges connected to these subgraphs to tend toward zero", how to initialize z is not described.
>
> Answer: The theoretical underpinnings of our proposed NM regularization method have been elaborated in Section 3.5 of the updated paper, including the initialization process for z.
>
> 6.he experimental results require detailed description to the experimental environment as in other referenced works
>
> Answer: We have expanded the description of the experimental environment in Section F of the appendix.
>
> 7.he results in section 4.2-4.4 are quite foreseeable, which bring minor information to the readers.
>
> Answer: In response to your feedback, Sections 4.2 and 4.3 have been revamped to showcase the results of parallel NMs on larger models, with previous content moved to the appendix to maintain focus.
>
> 8.The contents in appendix are redundant and also bring little information.
>
> Answer: To enhance the value of our appendix, we have included additional important information, such as a theoretical analysis of the motivation for reorganizing neural networks from a tree-like to a graph structure in Section C.
>
> We believe these revisions have significantly improved the clarity and depth of our paper. We appreciate your thorough review and are hopeful that our responses and revisions meet with your approval.
>
> Best regards,

---

> > ### Author Response · Authors · 2024-11-28
> >
> > Dear Respected Reviewer,
> >
> > We have added new experiments in the F.2 section of the appendix in the newly revised paper. These new experiments introduce a new dataset with thousands of features to assess the performance and efficiency of our method in comparison with traditional neural networks(NN). Here we try to conduct the experiments on a new, more complex dataset. Although, our approach does not outperform traditional NNs in terms of speed due to the increased number of computation steps we have incorporated. However, we have observed a significant improvement in accuracy.
> >
> > We acknowledge that NN may suffer from over fitting when too many nodes are added. Our method, while potentially slower, offers a substantial advantage in terms of accuracy when compared to the best results achieved by traditional NNs. Therefore, for tasks that prioritize higher accuracy over speed, our method stands as a strong candidate, particularly in fields such as medicine where precision is paramount.
> >
> > We have also noted that other existing methods, such as DEQ and OPTNET, have added extra computation steps to traditional NNs to improve accuracy with the cost of running time. However, as demonstrated in our experiments, our approach surpasses them in both performance and efficiency.
> >
> > Best regards

---

> ### Author Response · Authors · 2024-11-30
>
> Dear Respected Reviewer,
>
> Please give me a chance to address your first concern with additional clarification.
>
> I would like to explain  that our structure is not merely "flat." Within our framework, the structure is defined as a general graph, which allows for a more flexible and complex arrangement of neural modules. "Hierarchical structures" are also treated as neural modules in our approach, meaning our structure is not confined to a single level. If necessary, our framework can automatically organize into any hierarchical depth structure. A detailed theoretical explanation is provided in Section C of the appendix in our updated submission.
>
> Best regards,

---

> ### Author Response · Authors · 2024-12-03
> **The Essential Aspects of Our Framework**
>
> Dear Respected Reviewer,
>
> Here, please allow me the opportunity to explain the key aspects of our work.
>
> 1. First and foremost, we contend that any neural network (NN) structure can be conceptualized as a graph in geometric terms, which can be algebraically resolved by addressing a system of equations. The prevalent tree-like structures can also be regarded as a special case within this framework, which is predicated on an asynchronous approach to solving the system of equations, as depicted in Figure 11 of Section C in the appendix. The asynchronous nature of existing structures facilitates the solution of these equations and has enabled neural networks to be extensively applied across various scenarios. However, these structures often overlook the synchronous relationships among neurons that enable them to function cohesively. Consequently, we argue that existing structures only tap into a fraction of the potential of NNs. Our work aims to generalize the capabilities of NNs to their fullest extent, which has been theoretically explicated in Section C of the appendix. This forms the core motivation of our research.
>
> 2. We have fully generalized the neural network (NN) structure to a general graph which evolving to solve a system of equations. When the system of equations is synchronous and large, solving it can be computationally expensive, and the general graph structure would also become quite complex, introducing structural risk and potentially leading to overfitting. To address this, we approximate the general graph by retaining only the critical edges whose absolute weight are larger than parameter \gamma, and ignoring the other edges. This approximation simplifies the solution of the system of equations. With a larger \gamma, the approximation of the general graph will consist of several connected subgraphs, which we refer to as Neural Modules(NMs) in this paper, as depicted in Definition 1 of Section 1. This is the factorization of Neural Modules.
>
> 3. However, using the threshold \gamma to approximate the original general graph presents a challenge: the resulting Neural Modules (NMs) may be significantly unbalanced. There would always be one large neural module and many isolated neurons as small neural modules, which is illustrated in Figure 6 of Section 4.3. This imbalance leads to several issues:
>
> Performance: Isolated neurons are not utilized efficiently. Moreover, the large neural module introduces additional structural risks due to its complexity, which increases the likelihood of overfitting.
>
> Efficiency: As noted in the paper, the NMs are designed to be independent, allowing each Neural Module to be processed in parallel. This approach breaks down the original general graph into smaller NMs. We have observed that the overall efficiency of the solution process is contingent upon the size of the largest NM; if one NM is excessively large, the overall efficiency cannot be optimized effectively.
>
> Given these considerations, we have introduced NM regularization to facilitate the creation of balanced NMs. This regularization improves performance by mitigating overfitting through the reduction of structural risks and enhances efficiency by making the modules more amenable to parallel computing. As illustrated in section 3.5 and proved in section 4.1, section 4.2 and section 4.3
>
> 4. NM regularization is designed to integrate the scale of the Neural Module (NM) into the standard regularization process. The aim is to penalize large neural modules while simultaneously encouraging the growth of smaller neural modules. Equation 8 outlines NM regularization in the forward process. For the backward process, the derivatives of Equation 8 can be computed accordingly. If we set \alpha = 1, Equations 9 and 10 revert to the traditional solution of L1 and L2 regularization, respectively (appropriate references should be included here), as illustrated in section 3.5.
>
> 5. Throughout the entire learning process, the parameter \gamma is solely utilized to generate Neural Modules (NMs) for solving the equations. The update process encompasses all edges, meaning that the absolute value of any edge has the potential to surpass the parameter \gamma and thus create new NMs. The efficacy of NM regularization is demonstrated in Section 4.3 and the process of updating the weights is illustrated in section G of the appendix.
>
> These are the essential points of our approach. I hope this will be helpful. Thank you very much.
>
> Best regards.

---

### Official Review · Reviewer_8UEk · 2024-11-03

**Soundness:** 2
**Presentation:** 2
**Contribution:** 3
**Rating:** 5
**Confidence:** 3

**Summary:**

This paper proposes Neural Modules, a graph-based framework for organizing neural units within neural networks that aims to go beyond the traditional constrained structure of neural networks (NNs). Unlike the tree-like structure of traditional NNs which does not allow neurons to establish connections across the same layer, Neural Modules defines a graph-based structure that allows all neural units to communicate with all other units, allowing for more collaborative learning. Performance across three datasets from the UCI data repository are reported, showing that Neural Modules achieves lower error rates compared to other architectures and faster runtime compared to the DEQ model.

**Strengths:**

1. The discussion about traditional tree-like neural network tructure and the motivation behind redesigning the connectivity pattern behind neurons is well-written and clearly stated. A better organization of neurons could potentially allow for more efficient learning closer to that of real biological neural networks.
2. The methodology provides a formulation for the forward and backward pass of the Neural Modules architecture.
3. The authors show performance comparisons when varying the NM regularization parameter, and show a runtime comparison against one baseline model.

**Weaknesses:**

1. Performance of Neural Modules with several hundred nodes is reported on several datasets, however there are many other datasets with thousands of input features, requiring more nodes. The results would be stronger with benchmarking results on other datasets with more nodes in the graph, to demonstrate scalability to large graphs of neurons and more diverse datasets.
2. The running time reported for Neural Modules in Figure 3 lacks units, which makes it difficult to gauge what the relative improvement in runtime is against DEQ. Additionally, runtime is not compared to regular NNs, making it unclear if there are runtime disadvantages for Neural Modules versus traditional NNs.

**Questions:**

1. In Section 3.1, the authors mention that N0 denotes the “nodes for the first layer and the input variables feed into the first layer.” Does this indicate that N0 is the data which is fed into the first layer? The wording surrounding the usage of the words “input variables” is slightly confusing, however it is clear that the parameters of the model are represented by edges within the structure.
2. How well does this graph-structured processing work on modern GPU infrastructure? An ablation study of the runtime of Neural Modules versus traditional modules would provide more clarity on whether Neural Modules is comparable in runtime to traditional NNs on modern compute hardware.
3. How well does the Neural Modules framework scale to large neural layers with many nodes within a single layer, given that graph connectivity would introduce significant dependencies and computations between individual neurons? The authors propose NM regularization as a way of introducing sparsity in the connections in the graph to reduce the amount of computation, however it is unclear whether the model can optimally learn connections between nodes given the constraint. Figure 3 shows performance relative to different sparsity levels with 100 nodes on three datasets, however 100 nodes is quite small for many datasets which may have thousands of input features. A scalability test with more input nodes would strengthen the case that the framework is scalable to many nodes in a graph. Regarding the runtime reported in the fourth subplot of Figure 3, it would help to include units and more descriptive axes labels to help interpretation of how much runtime is benefitting from the sparsity constraint.
4. In several sections, there are typos or missing elements; in Figure 3, the subplots are not labeled with dataset labels, making it difficult to tell which dataset performance is being reported on in each subplot. In Section 3, there are some instances of unclear phrasing (e.g. “In real-world applications, efficiency can be optimized.” in Section 3.3).

---

> ### Author Response · Authors · 2024-11-24
>
> Dear Respected Reviewer:
>
> Thank you for your valuable feedback. I am pleased to provide detailed responses to your queries and to confirm that we have made the necessary revisions to our manuscript.
>
> We have been diligently conducting experiments further, particularly focusing on neural networks (NNs) with a larger number of nodes. The parallel methods discussed in our paper have yielded intriguing results, which are now detailed in Section 4.2 of the revised paper. The associated code for the parallel version has also been included in the supplementary material submitted alongside the new paper.  The code is for our experiments on the Postural Transitions dataset.
>
> Our Neural Modules (NMs) framework takes into account the relationships among nodes at the same level and their influence on the values of nodes in the preceding levels. We have chosen to compare our approach with DEQ, which also considers intra-level node relationships. Traditional NNs often overlook these relationships, which is why we have not included them in our comparisons.
>
> Further clarification on the rationale for transitioning from a tree-like to a graph structure in neural networks. This new theoretical explanation is now detailed in Section C of the appendix in our updated submission.
>
> In response to your questions:
>
> 1.In Section 3.1, the authors mention that N0 denotes the “nodes for the first layer and the input variables feed into the first layer.” Does this indicate that N0 is the data which is fed into the first layer? The wording surrounding the usage of the words “input variables” is slightly confusing, however it is clear that the parameters of the model are represented by edges within the structure.
>
> Answer：This has been revised for clarity in the newly submitted paper.
>
> 2.How well does this graph-structured processing work on modern GPU infrastructure? An ablation study of the runtime of Neural Modules versus traditional modules would provide more clarity on whether Neural Modules is comparable in runtime to traditional NNs on modern compute hardware.
>
> Answer：To address your inquiry about the performance of our graph-structured processing on modern GPU infrastructure, we have conducted additional experiments with larger parameters on GPUs. These results are presented in Section 4.2 of the revised paper, with detailed GPU specifications provided in the appendix. Our analysis aims to elucidate the applicability of our framework to large, realistic parameter sizes.
>
> 3.How well does the Neural Modules framework scale to large neural layers with many nodes within a single layer, given that graph connectivity would introduce significant dependencies and computations between individual neurons? The authors propose NM regularization as a way of introducing sparsity in the connections in the graph to reduce the amount of computation, however it is unclear whether the model can optimally learn connections between nodes given the constraint. Figure 3 shows performance relative to different sparsity levels with 100 nodes on three datasets, however 100 nodes is quite small for many datasets which may have thousands of input features. A scalability test with more input nodes would strengthen the case that the framework is scalable to many nodes in a graph. Regarding the runtime reported in the fourth subplot of Figure 3, it would help to include units and more descriptive axes labels to help interpretation of how much runtime is benefitting from the sparsity constraint.
>
> Answer：Concerning the scalability of our Neural Modules framework to large neural models with numerous nodes, we have included an expanded discussion and results in Section 4.2 of the revised paper. We have also enhanced related Figure to display performance with a significantly larger number of nodes across three datasets. This should provide a stronger demonstration of our framework's scalability, especially with regards to the runtime benefits of the regularization.
>
> 4.In several sections, there are typos or missing elements; in Figure 3, the subplots are not labeled with dataset labels, making it difficult to tell which dataset performance is being reported on in each subplot. In Section 3, there are some instances of unclear phrasing (e.g. “In real-world applications, efficiency can be optimized.” in Section 3.3).
>
> Answer：We have carefully reviewed and corrected the typos and missing elements you pointed out, including labeling the subplots in Figures with dataset identifiers.
>
> We believe these revisions have significantly improved the manuscript and addressed your concerns. We appreciate your thorough review.
>
> Best regards

---

> > ### Comment · Reviewer_8UEk · 2024-11-26
> >
> > Thank you for the response. The authors have added additional experiments with GPU hardware, which addresses some of my concerns. However, the baselines compared to Neural Modules is inconsistent in Figure 4a and b in the updated paper, which makes it hard to judge the runtime of Neural Modules versus baseline models. Overall, I will increase my score to a 5 for the authors efforts in running GPU experiments during the rebuttal process.

---

> > > ### Author Response · Authors · 2024-11-28
> > >
> > > Dear Respected Reviewer,
> > >
> > > Thank you very much for your consideration and the additional points awarded.
> > >
> > > The use of different baselines in Figures 4a and 4b is intentional due to the varying scales of nodes presented. Figure 4a, which covers nodes up to 300, utilizes a similar model, DEQ, as a baseline for comparison. In contrast, Figure 4b deals with nodes up to 3000, where DEQ becomes infeasible as it would take an excessive amount of time—potentially a month—to run on our computing resources. Therefore, for Figure 4b, we have chosen to compare our method on GPU with parallel computation to our standard NMs. This comparison aims to demonstrate that the parallel approach on contemporary hardware can significantly expand the potential application scope.
> > >
> > > Warm regards,

---

> > > > ### Author Response · Authors · 2024-11-28
> > > >
> > > > Dear Respected Reviewer,
> > > >
> > > > We have added new experiments in the F.2 section of the appendix in the newly revised paper. These new experiments introduce a new dataset with thousands of features to assess the performance and efficiency of our method in comparison with traditional neural networks(NN). Here we try to conduct the experiments on a new, more complex dataset.  Although, our approach does not outperform traditional NNs in terms of speed due to the increased number of computation steps we have incorporated. However, we have observed a significant improvement in accuracy.
> > > >
> > > > We acknowledge that NN may suffer from over fitting when too many nodes are added. Our method, while potentially slower, offers a substantial advantage in terms of accuracy when compared to the best results achieved by traditional NNs. Therefore, for tasks that prioritize higher accuracy over speed, our method stands as a strong candidate, particularly in fields such as medicine where precision is paramount.
> > > >
> > > > We have also noted that other existing methods, such as DEQ and OPTNET, have added extra computation steps to traditional NNs to improve accuracy with the cost of running time. However, as demonstrated in our experiments, our approach surpasses them in both performance and efficiency.
> > > >
> > > > Best regards

---

> ### Author Response · Authors · 2024-12-03
> **The Essential Aspects of Our Framework**
>
> Dear Respected Reviewer,
>
> Here, please allow me the opportunity to explain the key aspects of our work.
>
> 1. First and foremost, we contend that any neural network (NN) structure can be conceptualized as a graph in geometric terms, which can be algebraically resolved by addressing a system of equations. The prevalent tree-like structures can also be regarded as a special case within this framework, which is predicated on an asynchronous approach to solving the system of equations, as depicted in Figure 11 of Section C in the appendix. The asynchronous nature of existing structures facilitates the solution of these equations and has enabled neural networks to be extensively applied across various scenarios. However, these structures often overlook the synchronous relationships among neurons that enable them to function cohesively. Consequently, we argue that existing structures only tap into a fraction of the potential of NNs. Our work aims to generalize the capabilities of NNs to their fullest extent, which has been theoretically explicated in Section C of the appendix. This forms the core motivation of our research.
>
> 2. We have fully generalized the neural network (NN) structure to a general graph which evolving to solve a system of equations. When the system of equations is synchronous and large, solving it can be computationally expensive, and the general graph structure would also become quite complex, introducing structural risk and potentially leading to overfitting. To address this, we approximate the general graph by retaining only the critical edges whose absolute weight are larger than parameter \gamma, and ignoring the other edges. This approximation simplifies the solution of the system of equations. With a larger \gamma, the approximation of the general graph will consist of several connected subgraphs, which we refer to as Neural Modules(NMs) in this paper, as depicted in Definition 1 of Section 1. This is the factorization of Neural Modules.
>
> 3. However, using the threshold \gamma to approximate the original general graph presents a challenge: the resulting Neural Modules (NMs) may be significantly unbalanced. There would always be one large neural module and many isolated neurons as small neural modules, which is illustrated in Figure 6 of Section 4.3. This imbalance leads to several issues:
>
> Performance: Isolated neurons are not utilized efficiently. Moreover, the large neural module introduces additional structural risks due to its complexity, which increases the likelihood of overfitting.
>
> Efficiency: As noted in the paper, the NMs are designed to be independent, allowing each Neural Module to be processed in parallel. This approach breaks down the original general graph into smaller NMs. We have observed that the overall efficiency of the solution process is contingent upon the size of the largest NM; if one NM is excessively large, the overall efficiency cannot be optimized effectively.
>
> Given these considerations, we have introduced NM regularization to facilitate the creation of balanced NMs. This regularization improves performance by mitigating overfitting through the reduction of structural risks and enhances efficiency by making the modules more amenable to parallel computing. As illustrated in section 3.5 and proved in section 4.1, section 4.2 and section 4.3
>
> 4. NM regularization is designed to integrate the scale of the Neural Module (NM) into the standard regularization process. The aim is to penalize large neural modules while simultaneously encouraging the growth of smaller neural modules. Equation 8 outlines NM regularization in the forward process. For the backward process, the derivatives of Equation 8 can be computed accordingly. If we set \alpha = 1, Equations 9 and 10 revert to the traditional solution of L1 and L2 regularization, respectively (appropriate references should be included here), as illustrated in section 3.5.
>
> 5. Throughout the entire learning process, the parameter \gamma is solely utilized to generate Neural Modules (NMs) for solving the equations. The update process encompasses all edges, meaning that the absolute value of any edge has the potential to surpass the parameter \gamma and thus create new NMs. The efficacy of NM regularization is demonstrated in Section 4.3 and the process of updating the weights is illustrated in section G of the appendix.
>
> These are the essential points of our approach. I hope this will be helpful.  Thank you very much.
>
> Best regards.

---

### Official Review · Reviewer_ij2R · 2024-11-04

**Soundness:** 3
**Presentation:** 3
**Contribution:** 2
**Rating:** 5
**Confidence:** 3

**Summary:**

This paper contends that current tree-like/hierarchical structures of neural networks are insufficiently expressive compared to examples of biological neural networks, proposes that networks of arbitrary directed graph structure (where "later" nodes can influence "earlier" ones) might get us access to levels of expressivity not possible with existing structures. However, they recognize that, because this model structure necessitates solving a system of equations in the forward pass, it scales poorly with the size of each system of equations being solved (which in this case is the size of the fully connected subcomponent being solved). Therefore, they formulate a regularizer (which they call NM regularization) which penalizes nodes for being connected to large subcomponents within the graph. This means that nodes will tend to have their edges with larger subgraphs decay to zero, which reduces the size of the subgraphs. They also implement threshold-based sparsification, only including equations in the system of equations to be solved if they have a weight > lambda to the subcomponent

**Strengths:**

- The paper provides a seemingly practical way of scaling computation graphs without scaling the systems of equations to be solved to an impractical degree, while still staying within the paradigm of soft, gradient-based learning, and not needing to know the network structure ahead of time
- The paper is well-presented and clear to read

**Weaknesses:**

- The paper didn't provide any estimates for how computational cost would scale for more plausibly-sized models (which I think could have been done with some estimates of different regularization parameters and size k without necessarily having to run those experiments)
- The paper did not necessarily make an argument for whether certain _kinds_ of dataset benefit from this problem structure or require it to be solved well, which would make the argument for this computationally costly procedure more compelling. Instead it just tested the approach on some generic ML datasets.

**Questions:**

- Is there any story for how this could scale to realistic sizes of parameters (as far as I saw 300 parameters was the largest size shown)?
- What kinds of problems do you believe this technique would be specifically high-value for?

---

> ### Author Response · Authors · 2024-11-24
>
> Dear Respected Reviewer：
>
> Thank you for your insightful comments and questions. Further clarification on the rationale for transitioning from a tree-like to a graph structure in neural networks. This new theoretical explanation is now detailed in Section C of the appendix in our updated submission. Below are my detailed responses to your inquiries.
>
> 1，Is there any story for how this could scale to realistic sizes of parameters (as far as I saw 300 parameters was the largest size shown)?
>
> Answer：We have expanded our experiments. As detailed in Section 4.2 of the revised manuscript, we have successfully scaled our framework to handle more substantial parameter sets. The parallel methods discussed in the paper have been instrumental in these experiments, and the results are promising. The corresponding code for the parallel version is included in the supplementary material submitted with the new paper.  The code is for our experiments on the Postural Transitions dataset. Our analysis of these results provides a clear explanation of how our framework can be  effectively applied to larger, more realistic parameter sizes.
>
> In our expanded experiments, we utilized a network with 3000 nodes, resulting in a 3000 by 3000 parameter matrix, amounting to millions of parameters. Should we have access to more powerful computing resources, our framework could be scaled to accommodate billions of parameters.
>
> 2.What kinds of problems do you believe this technique would be specifically high-value for?
>
> Answer： As for the types of problems where our technique would be particularly valuable, we believe our framework, which aims to enhance the structure of neural networks (NNs), has broad applicability. To demonstrate this, we have conducted experiments on a variety of real-world machine-learning datasets spanning genetics, physics, and kinematics, among others. The encouraging results from these experiments suggest that our approach can indeed be generalized across different domains.
>
> In response to your question， consider the introduction of new parameters for our structure, we think that our method is better suited for more complex problems with enough training data， as the additional parameters can increase the risk of overfitting if not properly managed with sufficient data.
>
> We hope these responses address your questions and provide a clearer understanding of our research and its implications.
>
> Best regards

---

> > ### Comment · Reviewer_ij2R · 2024-11-26
> >
> > Thanks to the authors for their engagement with my concerns.
> >
> > - While the added experiments up to 3000 nodes are a step in the right direction, it is still quite far from modern methods, and it's still unclear how to ground this increased runtime vs performance tradeoff without knowing the runtime characteristics of more traditional NNs at this scale. The question I would want to get a positive answer to is: how do these methods compare to traditional NNs when controlling for compute or runtime, since, while NNs perform slightly worse on the datasets tested, I assume that NNs are also substantially faster, and it would be useful to answer questions like "how big of a standard NN can you train in the same runtime that you can train a NM of size X"
> > - While there are multiple datasets shown in this paper, it's still not clear whether there are problems that a big enough NN fundamentally will have issues solving, and that a NM will solve successfully. To the point above: can I just train a bigger NN at the same runtime cost to match the performance of a NM. To answer that question, looking more closely at the results, I notice that for two out of the three datasets tested, the 300 node has higher error than the 200 node for NMs, and for all 3 datasets the standard NN has higher error on the 300 node relative to the 200 node. This raises concerns about overfitting, and the simplicity of these datasets, if they are saturating at this small a number of parameters, and makes me worry that these datasets are too simple to provide experimental signal that we should expect to generalize elsewhere

---

> ### Author Response · Authors · 2024-11-28
> **Paper  Newly Revised**
>
> Dear Respected Reviewer,
>
> I would like to begin by extending my sincere apologies for not fully grasping your constructive feedback on my initial submission. I appreciate the opportunity to address your valuable concerns within the tight one-day deadline.
>
> In response to your valuable input, we have added new experiments in the F.2 section of the appendix in the newly revised paper. These new experiments introduce a new dataset with thousands of features to assess the performance and efficiency of our method in comparison with traditional neural networks(NN). Frankly, our approach does not outperform traditional NNs in terms of speed due to the increased number of computation steps we have incorporated. However, we have observed a significant improvement in accuracy.
>
> We acknowledge that NN can suffer from overfitting when too many nodes are added. Our method, while potentially slower, offers a substantial advantage in terms of accuracy when compared to the best results achieved by traditional NNs. Therefore, for tasks that prioritize higher accuracy over speed, our method stands as a strong candidate, particularly in fields such as medicine where precision is paramount.
>
> We have also noted that other existing methods, such as DEQ and OPTNET, have added extra computation steps to traditional NNs to improve accuracy with the cost of running time. However, as demonstrated in our experiments, our approach surpasses them in both performance and efficiency.
>
> I apologize for placing the crucial additional experiments in the appendix. Given the one-day constraint, we did not have sufficient time to integrate these results into the main body of the paper. Our focus was on conducting the experiments on a new, more complex dataset using the renting cloud server resources. Despite this, we believe that our efforts are worthwhile in advancing our work based on your insightful comments.
>
> Thank you very much for your comments.
>
> Yours sincerely

---

> ### Author Response · Authors · 2024-12-03
> **The Essential Aspects of Our Framework**
>
> Dear Respected Reviewer,
>
> Here, please allow me the opportunity to explain the key aspects of our work.
>
> 1. First and foremost, we contend that any neural network (NN) structure can be conceptualized as a graph in geometric terms, which can be algebraically resolved by addressing a system of equations. The prevalent tree-like structures can also be regarded as a special case within this framework, which is predicated on an asynchronous approach to solving the system of equations, as depicted in Figure 11 of Section C in the appendix. The asynchronous nature of existing structures facilitates the solution of these equations and has enabled neural networks to be extensively applied across various scenarios. However, these structures often overlook the synchronous relationships among neurons that enable them to function cohesively. Consequently, we argue that existing structures only tap into a fraction of the potential of NNs. Our work aims to generalize the capabilities of NNs to their fullest extent, which has been theoretically explicated in Section C of the appendix. This forms the core motivation of our research.
>
> 2. We have fully generalized the neural network (NN) structure to a general graph which evolving to solve a system of equations. When the system of equations is synchronous and large, solving it can be computationally expensive, and the general graph structure would also become quite complex, introducing structural risk and potentially leading to overfitting. To address this, we approximate the general graph by retaining only the critical edges whose absolute weight are larger than parameter \gamma, and ignoring the other edges. This approximation simplifies the solution of the system of equations. With a larger \gamma, the approximation of the general graph will consist of several connected subgraphs, which we refer to as Neural Modules(NMs) in this paper, as depicted in Definition 1 of Section 1. This is the factorization of Neural Modules.
>
> 3. However, using the threshold \gamma to approximate the original general graph presents a challenge: the resulting Neural Modules (NMs) may be significantly unbalanced. There would always be one large neural module and many isolated neurons as small neural modules, which is illustrated in Figure 6 of Section 4.3. This imbalance leads to several issues:
>
> Performance: Isolated neurons are not utilized efficiently. Moreover, the large neural module introduces additional structural risks due to its complexity, which increases the likelihood of overfitting.
>
> Efficiency: As noted in the paper, the NMs are designed to be independent, allowing each Neural Module to be processed in parallel. This approach breaks down the original general graph into smaller NMs. We have observed that the overall efficiency of the solution process is contingent upon the size of the largest NM; if one NM is excessively large, the overall efficiency cannot be optimized effectively.
>
> Given these considerations, we have introduced NM regularization to facilitate the creation of balanced NMs. This regularization improves performance by mitigating overfitting through the reduction of structural risks and enhances efficiency by making the modules more amenable to parallel computing. As illustrated in section 3.5 and proved in section 4.1, section 4.2 and section 4.3
>
> 4. NM regularization is designed to integrate the scale of the Neural Module (NM) into the standard regularization process. The aim is to penalize large neural modules while simultaneously encouraging the growth of smaller neural modules. Equation 8 outlines NM regularization in the forward process. For the backward process, the derivatives of Equation 8 can be computed accordingly. If we set \alpha = 1, Equations 9 and 10 revert to the traditional solution of L1 and L2 regularization, respectively (appropriate references should be included here), as illustrated in section 3.5.
>
> 5. Throughout the entire learning process, the parameter \gamma is solely utilized to generate Neural Modules (NMs) for solving the equations. The update process encompasses all edges, meaning that the absolute value of any edge has the potential to surpass the parameter \gamma and thus create new NMs. The efficacy of NM regularization is demonstrated in Section 4.3 and the process of updating the weights is illustrated in section G of the appendix.
>
> These are the essential points of our approach. I hope this will be helpful. Thank you very much.
>
> Best regards.

---

### Official Review · Reviewer_ZySP · 2024-11-04

**Soundness:** 2
**Presentation:** 3
**Contribution:** 2
**Rating:** 6
**Confidence:** 3

**Summary:**

This paper introduces the idea of replacing feed-forward neural networks with densely connected networks of neural-units that allow all-to-all communication. It also defines the idea of a "Neural Module", a locally connected subgraph is such all-to-all networks.

**Strengths:**

This paper does a good job of introducing and motivating the idea of graph-based neural networks and Neural Modules. Presentation of results on multiple datasets helps gain confidence in the ability for this idea to positively impact current research. The work opens up new avenues of research in the fundamental design of neural networks.

**Weaknesses:**

The paper is missing certain details about the method - such as what algorithm is used to solve the set of simultaneous equations in the forward process, and how neural-modules are identified and factorized.

**Questions:**

I have 2 major questions that effect my score. In the order of importance:

1. The paper mentions ability of a neural-graph $\mathcal{G}$ to be separated into Neural Modules for faster computation by allowing parallel computation, but the method for such factorization or parallel computation is never presented. This I believe is the most important detail that this paper should provide as it allows the realization of neural modules effectively making such general neural-graphs computationally feasible.
2. It is unclear as to how the introduction of regularization in the forward process (Eqn. 6) directly achieves promotion of sparsity in the network. Typically, such regularization is done in the optimization step after computing the gradients (eg. L2 regularization of weights in a feed-forward neural network). Any explaination of this phenomenon, or ablation experiments proving this as a valid approach in comparison to traditional regularization in optimization step, would be helpful.


**Small suggestions to improve the paper**:
1. Line 209: "Existing numerical methods can effectively solve these above equations." - It would be good to mention which methods you used.
2. Figure 3: The x-axis of all plots is mentioned as "parameter", however it is $\alpha$ for the first 3 plots, and "number of nodes" for the last one. The figure should be updated with correct x-axis labels to help readers. Additionally, the different plots should be labeled as A, B, C, D, so that they can be referred in the text as "Fig. 3A shows ...", instead of "first plot of figure 3 ..."
3. I believe the symbol $f^{-1}$ is incorrectly used in the paper to denote the derivative of the function $f$. This is the symbol is generally used for "inverse function for $f$", and the standard was derivative is $f'$.

---

> ### Author Response · Authors · 2024-11-24
>
> Dear Respected Reviewer：
>
> Thank you very much for your insightful comments and suggestions. We have carefully considered your feedback and have submitted a revised version of our paper that addresses your concerns. Specifically, we have made the following updates:
> In Section 3.3, we have expanded our explanation of the algorithm used to solve the set of simultaneous equations in the forward process. Additionally, in Section 3.5, we have detailed how neural modules are identified and factorized, which should clarify the method for parallel computation and factorization mentioned in the paper.
>
> Further clarification on the rationale for transitioning from a tree-like to a graph structure in neural networks. This new theoretical explanation is now detailed in Section C of the appendix in our updated submission.
>
> Here are my responses to your questions.
>
> 1.The paper mentions ability of a neural-graph to be separated into Neural Modules for faster computation by allowing parallel computation, but the method for such factorization or parallel computation is never presented. This I believe is the most important detail that this paper should provide as it allows the realization of neural modules effectively making such general neural-graphs computationally feasible.
>
> Answer：We have been conducting experiments, utilizing the parallel methods outlined in the new paper. The intriguing results of these experiments are now presented in Section 4.2 of the revised paper. Furthermore, the code for the parallel version has been included in the supplementary material submitted with the new version. The code is for our experiments on the Postural Transitions dataset.
>
> 2.It is unclear as to how the introduction of regularization in the forward process (Eqn. 6) directly achieves promotion of sparsity in the network. Typically, such regularization is done in the optimization step after computing the gradients (eg. L2 regularization of weights in a feed-forward neural network). Any explaination of this phenomenon, or ablation experiments proving this as a valid approach in comparison to traditional regularization in optimization step, would be helpful.
>
> Answer：Regarding your query about the introduction of regularization and its role in promoting network, we have provided a more detailed explanation in Section 3.5 of the revised paper. We hope this additional information will address your concerns and shed light on this aspect of our approach.
>
> We acknowledge your point about the typical placement of regularization in the optimization step and have included a discussion on why our method differs and its benefits. We have also added ablation experiments in the revised paper to validate our approach.
> We hope that these revisions will be helpful and address the questions you raised. We appreciate your time.
>
> Best regards.

---

> ### Author Response · Authors · 2024-11-24
>
> Thank you for your mall suggestions aimed at enhancing our paper. I am pleased to confirm that all your suggestions have been addressed and incorporated into the latest version of the manuscript we have submitted.

---

> > ### Author Response · Authors · 2024-11-28
> >
> > Dear Respected Reviewer,
> >
> > We have added new experiments in the F.2 section of the appendix in the newly revised paper. These new experiments introduce a new dataset with thousands of features to assess the performance and efficiency of our method in comparison with traditional neural networks(NN). Here we try to conduct the experiments on a new, more complex dataset.  Although, our approach does not outperform traditional NNs in terms of speed due to the increased number of computation steps we have incorporated. However, we have observed a significant improvement in accuracy.
> >
> > We acknowledge that NN may suffer from over fitting when too many nodes are added. Our method, while potentially slower, offers a substantial advantage in terms of accuracy when compared to the best results achieved by traditional NNs. Therefore, for tasks that prioritize higher accuracy over speed, our method stands as a strong candidate, particularly in fields such as medicine where precision is paramount.
> >
> > We have also noted that other existing methods, such as DEQ and OPTNET, have added extra computation steps to traditional NNs to improve accuracy with the cost of running time. However, as demonstrated in our experiments, our approach surpasses them in both performance and efficiency.
> >
> > Best regards

---

> ### Comment · Reviewer_ZySP · 2024-12-03
>
> I thank the authors for their response to my comments. However, I believe that neither of my major concerns has been fully addressed by the updates to the text.
>
> - Re: NM separability details - Although it is good to see that the authors have been able to increase the scale of the tested networks by employing parallel compute (GPUs), however, still no implementation details about the factorization and processing of the network into NMs has been added to the text. Since the idea of factorizing a graph into NMs is a major contribution of this paper, these details cannot be left for the readers/reviewers to gather from supplementary material (code).
>
> - Re: NM Regularization: I believe my concern has been _partly_ answered. I think the rows corresponding to "NMs&L1" in Table 1 and 2 follow the "conventional" regularization done in the optimization step, and they clearly perform worse in comparison to "NMs&NM". However, the motivation for NM regularization term being in the forward pass equation is still unclear to me. Eqn. 9 and 10 present "optimal value" of _something_ without ever setting up an optimization problem.
>
> I maintain my score. I believe this idea has high potential, however the current presentation/narrative is severely lacking rigor and motivation. A "complete" presentation of this idea would include algorithmic details about NM factorization (along with complexity analysis), and motivation for the proposed NM regularization (beyond just having higher performance).

---

> ### Author Response · Authors · 2024-12-03
> **The Essential Aspects of Our Framework**
>
> Dear Respected Reviewer,
>
> Thank you very much for your thorough review. First, I would like to apologize for any unclear points in our paper. We should incorporate more information, such as algorithmic details, into our manuscript. Here, please allow me the opportunity to explain the key aspects of our work that you have highlighted.
>
> 1. First and foremost, we contend that any neural network (NN) structure can be conceptualized as a graph in geometric terms, which can be algebraically resolved by addressing a system of equations. The prevalent tree-like structures can also be regarded as a special case within this framework, which is predicated on an asynchronous approach to solving the system of equations, as depicted in Figure 11 of Section C in the appendix. The asynchronous nature of existing structures facilitates the solution of these equations and has enabled neural networks to be extensively applied across various scenarios. However, these structures often overlook the synchronous relationships among neurons that enable them to function cohesively. Consequently, we argue that existing structures only tap into a fraction of the potential of NNs. Our work aims to generalize the capabilities of NNs to their fullest extent,  which has been theoretically explicated in Section C of the appendix. This forms the core motivation of our research.
>
> 2. We have fully generalized the neural network (NN) structure to a general graph which evolving to solve a system of equations. When the system of equations is synchronous and large, solving it can be computationally expensive, and the general graph structure would also become quite complex, introducing structural risk and potentially leading to overfitting. To address this, we approximate the general graph by retaining only the critical edges whose absolute weight are larger than parameter \gamma, and ignoring the other edges. This approximation simplifies the solution of the system of equations. With a larger \gamma, the approximation of the general graph will consist of several connected subgraphs, which we refer to as Neural Modules(NMs) in this paper, as depicted in Definition 1 of Section 1. This is the factorization of Neural Modules.
>
> 3. However, using the threshold \gamma to approximate the original general graph presents a challenge: the resulting Neural Modules (NMs) may be significantly unbalanced. There would always be one large neural module and many isolated neurons as small neural modules, which is illustrated in Figure 6 of Section 4.3. This imbalance leads to several issues:
>
> Performance: Isolated neurons are not utilized efficiently. Moreover, the large neural module introduces additional structural risks due to its complexity, which increases the likelihood of overfitting.
>
> Efficiency: As noted in the paper, the NMs are designed to be independent, allowing each Neural Module to be processed in parallel. This approach breaks down the original general graph into smaller NMs. We have observed that the overall efficiency of the solution process is contingent upon the size of the largest NM; if one NM is excessively large, the overall efficiency cannot be optimized effectively.
>
> Given these considerations, we have introduced NM regularization to facilitate the creation of balanced NMs. This regularization improves performance by mitigating overfitting through the reduction of structural risks and enhances efficiency by making the modules more amenable to parallel computing. As illustrated in section 3.5 and proved in section 4.1, section 4.2 and section 4.3
>
> 4. NM regularization is designed to integrate the scale of the Neural Module (NM) into the standard regularization process. The aim is to penalize large neural modules while simultaneously encouraging the growth of smaller neural modules. Equation 8 outlines NM regularization in the forward process. For the backward process, the derivatives of Equation 8 can be computed accordingly. If we set \alpha = 1, Equations 9 and 10 revert to the traditional solution of L1 and L2 regularization, respectively (appropriate references should be included here), as illustrated in section 3.5.
>
> 5. Throughout the entire learning process, the parameter \gamma is solely utilized to generate Neural Modules (NMs) for solving the equations. The update process encompasses all edges, meaning that the absolute value of any edge has the potential to surpass the parameter \gamma and thus create new NMs. The efficacy of NM regularization is demonstrated in Section 4.3 and the process of updating the weights is illustrated in Section G of the appendix.
>
> These are the essential points of our approach. I hope this will be helpful. Thank you very much, and we eagerly await your reassessment, if possible.
>
> Best regards.

---

### Author Response · Authors · 2024-11-30
**Significant Improvements in the Updated Manuscript**

Dear Respected Chairs and Reviewers,

I am writing to outline the significant improvements incorporated into the newly revised version of our paper.

1. Theoretical Expansion: We have introduced a comprehensive new theoretical explanation in Section C of the appendix, which elucidates the rationale for shifting from a tree-like to a graph structure in neural networks. This section not only highlights the limitations of traditional tree-like structures but also establishes how these can be considered a specific instance of our broader framework.

2. Extended Experiments on Larger Models: Building upon the proposals outlined in our paper, we have expanded our experimental scope to larger models, leveraging parallel computing on GPU as suggested in the paper. This extension ensures that our method is applicable to a wider range of real-world problems. The results, as detailed in Sections 4.2 and 4.3, are encouraging in both runtime performance and accuracy.  The original content from Sections 4.2 and 4.3 has been relocated to Sections F.1 and F.3 in the appendix in the new paper.
Furthermore，we have added new experiments as described in Section F.2 of the appendix. These experiments utilize new datasets with thousands of features to compare our framework's performance and efficiency against widely used traditional neural networks. In Section F.2, we primarily analyzed the application case of our framework.

3. More Detailed Introduction to The Process of NM Regularization: We also propose a comprehensive new theoretical explanation of NM Regularization and compare it with L1 and L2 regularization. It explains how neural-modules are identified and factorized in section 3.5 of the newly revised paper.

4. New Dataset Experiments: We have conducted new experiments on a new dataset with thousands of features to assess our method's performance and efficiency against widely used traditional neural networks (NNs). While our approach may not surpass traditional NNs in speed due to the additional computation steps, we have observed a notable enhancement in accuracy. Although potentially slower, our method offers a significant advantage in accuracy over the best results achieved by traditional NNs, making it a strong candidate for tasks where precision is more critical than speed, especially in fields such as medicine where accuracy is paramount. These new experiments are detailed in Section F.2 of the appendix in the revised paper.

5. The Relates with Reservoir Computing: We have enhanced the academic depth of our paper by incorporating an analysis of related work, with a focus on reservoir computing, in Section 2.

6. Addressing Reviewer Concerns: The issues highlighted by the reviewers have been addressed and resolved in the revised manuscript.

7. New Supplementary Material: We have included the new code for the parallel version on GPU in the newly submitted Supplementary Material. The code is for our experiments on the Postural Transitions dataset.




We believe these enhancements significantly contribute to the depth and applicability of our research. We appreciate your consideration.

Best regards.

---

> ### Author Response · Authors · 2024-12-03
> **The Essential Aspects of Our Framework**
>
> Dear Respected Reviewer,
>
> Here, please allow me the opportunity to explain the key aspects of our work.
>
> 1. First and foremost, we contend that any neural network (NN) structure can be conceptualized as a graph in geometric terms, which can be algebraically resolved by addressing a system of equations. The prevalent tree-like structures can also be regarded as a special case within this framework, which is predicated on an asynchronous approach to solving the system of equations, as depicted in Figure 11 of Section C in the appendix. The asynchronous nature of existing structures facilitates the solution of these equations and has enabled neural networks to be extensively applied across various scenarios. However, these structures often overlook the synchronous relationships among neurons that enable them to function cohesively. Consequently, we argue that existing structures only tap into a fraction of the potential of NNs. Our work aims to generalize the capabilities of NNs to their fullest extent, which has been theoretically explicated in Section C of the appendix. This forms the core motivation of our research.
>
> 2. We have fully generalized the neural network (NN) structure to a general graph which evolving to solve a system of equations. When the system of equations is synchronous and large, solving it can be computationally expensive, and the general graph structure would also become quite complex, introducing structural risk and potentially leading to overfitting. To address this, we approximate the general graph by retaining only the critical edges whose absolute weight are larger than parameter \gamma, and ignoring the other edges. This approximation simplifies the solution of the system of equations. With a larger \gamma, the approximation of the general graph will consist of several connected subgraphs, which we refer to as Neural Modules(NMs) in this paper, as depicted in Definition 1 of Section 1. This is the factorization of Neural Modules.
>
> 3. However, using the threshold \gamma to approximate the original general graph presents a challenge: the resulting Neural Modules (NMs) may be significantly unbalanced. There would always be one large neural module and many isolated neurons as small neural modules, which is illustrated in Figure 6 of Section 4.3. This imbalance leads to several issues:
>
> Performance: Isolated neurons are not utilized efficiently. Moreover, the large neural module introduces additional structural risks due to its complexity, which increases the likelihood of overfitting.
>
> Efficiency: As noted in the paper, the NMs are designed to be independent, allowing each Neural Module to be processed in parallel. This approach breaks down the original general graph into smaller NMs. We have observed that the overall efficiency of the solution process is contingent upon the size of the largest NM; if one NM is excessively large, the overall efficiency cannot be optimized effectively.
>
> Given these considerations, we have introduced NM regularization to facilitate the creation of balanced NMs. This regularization improves performance by mitigating overfitting through the reduction of structural risks and enhances efficiency by making the modules more amenable to parallel computing. As illustrated in section 3.5 and proved in section 4.1, section 4.2 and section 4.3
>
> 4. NM regularization is designed to integrate the scale of the Neural Module (NM) into the standard regularization process. The aim is to penalize large neural modules while simultaneously encouraging the growth of smaller neural modules. Equation 8 outlines NM regularization in the forward process. For the backward process, the derivatives of Equation 8 can be computed accordingly. If we set \alpha = 1, Equations 9 and 10 revert to the traditional solution of L1 and L2 regularization, respectively (appropriate references should be included here), as illustrated in section 3.5.
>
> 5. Throughout the entire learning process, the parameter \gamma is solely utilized to generate Neural Modules (NMs) for solving the equations. The update process encompasses all edges, meaning that the absolute value of any edge has the potential to surpass the parameter \gamma and thus create new NMs. The efficacy of NM regularization is demonstrated in Section 4.3 and the process of updating the weights is illustrated in section G of the appendix.
>
> These are the essential points of our approach. I hope this will be helpful. Thank you very much.
>
> Best regards.

---

### Meta-Review · Area_Chair_6h9m · 2024-12-23

**Metareview:**

This paper introduces a synchronous graph-based structure to replace standard feedforward networks. This allows for communication between features within a layer and across different subgraphs called “neural modules”.

The reviewers were enthusiastic about the overall idea but raised a number of concerns about: (i) how the partitioning of subgraphs into neural modules was achieved, (ii) the added complexity and sensitivity to this partitioning procedure, and (iii) justification for their regularization strategy. While the authors did provide some answers to these questions, the reviewers' concerns were not fully addressed. In the end, the reviewers agreed that the paper needs more work before it will be ready for publication.

**Additional Comments On Reviewer Discussion:**

The authors made a number of attempts to address the reviewer concerns, including a new theoretical justification of the regularization procedure, and extended results. However, the reviewers noted that their core concerns were not fully addressed by the revisions.

---

### Decision · Program_Chairs · 2025-01-22

Reject